# Structural convergence and water-mediated substrate mimicry enable broad neuraminidase inhibition by human antibodies

Julia Lederhofer [1,7], Andrew J. Borst [2,3,7], Lam Nguyen[1],
Rebecca A. Gillespie [1], Connor J. Williams[4], Emma L. Walker [4], Julie E. Raab[1],
Christina Yap[1], Daniel Ellis[2,3,5], Adrian Creanga [1], Hyon-Xhi Tan[6], Thi H. T. Do[6],
Michelle Ravichandran [1], Adrian B. McDermott [1], Valerie Le Sage[4],
Sarah F. Andrews [1], Barney S. Graham[1], Adam K. Wheatley [6],
Douglas S. Reed [4], Neil P. King [2,3] ✉ & Masaru Kanekiyo [1] ✉

Influenza has been responsible for multiple global pandemics and seasonal epidemics and claimed millions of lives. The imminent threat of a panzootic outbreak of avian influenza H5N1 virus underscores the urgent need for pandemic preparedness and effective countermeasures, including monoclonal antibodies (mAbs). Here, we characterize human mAbs that target the highly conserved catalytic site of viral neuraminidase (NA), termed NCS mAbs, and the molecular basis of their broad specificity. Cross-reactive NA-specific B cells were isolated by using stabilized NA probes of non-circulating subtypes. We found that NCS mAbs recognized multiple NAs of influenza A as well as influenza B NAs and conferred prophylactic protections in mice against H1N1, H5N1, and influenza B viruses. Cryo-electron microscopy structures of two NCS mAbs revealed that they rely on structural mimicry of sialic acid, the substrate of NA, by coordinating not only amino acid side chains but also water molecules, enabling inhibition of NA activity across multiple influenza A and B viruses, including avian influenza clade 2.3.4.4b H5N1 viruses. Our results provide a molecular basis for the broad reactivity and inhibitory activity of NCS mAbs targeting the catalytic site of NA through substrate mimicry.

Recurrent panzootic outbreaks of highly pathogenic avian influenza (HPAI) virus, such as by clade 2.3.4.4b H5N1 virus[1–5], pose a significant global health threat. In response, there is an urgent need for pandemic preparedness, particularly in identifying potential medical countermeasures effective against HPAI viruses such as broadly cross-protective monoclonal antibodies (mAbs). Most early antibody discovery efforts centered on targeting highly conserved epitopes on the viral hemagglutinin (HA) to block either viral attachment or membrane

fusion and thereby broadly neutralize a wide range of influenza viruses[6]. Several broadly neutralizing antibodies (bnAbs) targeting the HA stem domain have been advanced to human clinical trials with mixed results[7,8]. A key challenge with these antibodies is their relatively low potency, especially against influenza A viruses encompassing group 2 HA subtypes. For example, VIR-2482, one of the broadest and most potent HA stem-targeting bnAbs, recently failed to meet its efficacy endpoint when tested as a prophylaxis in a phase 2 clinical

trial[9–11]. Such results motivate the exploration of alternative strategies that focus on other viral targets.

Influenza neuraminidase (NA) is a glycoprotein present on the virion surface that facilitates nascent virus egress by cleaving sialosides from cell surface glycoproteins and glycolipids[12]. NA is a type 2 integral membrane protein and forms a homotetramer, with each protomer containing a catalytic site[12,13] that binds and hydrolyzes sialic acids. Indeed, inhibiting the catalytic activity of NA is an established strategy for limiting viral propagation[12,14]. Various NA inhibitors, such as oseltamivir and zanamivir, have demonstrated clinical efficacy by targeting the highly conserved NA catalytic pocket in diverse influenza A and B viral strains. The catalytic site has also been a prime target of broadly cross-reactive antibodies. Stadlbauer et al. identified a set of mAbs including 1G01 in a human subject that target the catalytic site and possess exceptional breadth of reactivity and protection, recapitulating the broad spectrum of licensed NA inhibitors[15]. There are a few other mAbs that have similar breadth and epitope specificity, including DA03E17[16] and FNI9[17]. Like 1G01, these mAbs target the conserved catalytic pocket and all inhibit NA enzymatic activity across multiple influenza A and B viruses[15–17].

1G01 features an abnormally long complementarity determining region 3 of the heavy chain (CDRH3) that penetrates deeply into the catalytic pocket, inhibiting enzymatic function without mimicking the natural substrate[15]. In contrast, FNI9 possesses a shorter CDRH3 and forms molecular interactions that closely mimic sialic acid binding, leading to enhanced potency compared to traditional inhibitors like oseltamivir and 1G01[17]. The structural basis for NA recognition and inhibition by DA03E17 is similar to these mAbs and is described in detail in the accompanying manuscript by Jo et al. These antibodies all represent promising candidates for influenza prophylaxis, offering broad reactivity and protection against various influenza A and B viruses. In addition, FNI9 was shown to have a synergistic effect in inhibiting viruses in vitro when combined with an HA stem-targeting bnAb[17], suggesting the utility of NA catalytic site-targeting mAbs in combination with other bnAbs to simultaneously target multiple viral sites of vulnerability to achieve broader breadth and higher potency.

Conventionally, recombinant NA proteins often suffer from structural instability, particularly in subtypes like N1, which limits their use as reliable probes for antibody discovery and structural characterization, particularly cryo-electron microscopy (cryoEM) studies[18]. To address these challenges, we recently developed a series of computationally-designed stabilized neuraminidase proteins (sNAps)[18]. These sNAps were designed to remain in the closed tetrameric conformation that is representative of the NA conformation on the surface of the virus, providing more biologically relevant molecular tools for characterizing antibody-antigen interactions. As a result, these sNAps have enabled the robust in vitro characterization of antibodies toward NA and facilitated structure determination of NA tetramers by cryoEM[18].

Here, we utilize sNAps to identify and characterize a set of broadly cross-reactive mAbs targeting N1 and other group 1 subtype NAs, termed NCS.1.x mAbs. Near-atomic resolution cryoEM analyses revealed that two mAbs, NCS.1 and NCS.1.1, leverage a conserved Asp-Arg motif within their CDRH3 loops, employing a lock-and-key binding modality supported by water molecules along the NA interface to precisely mimic sialic acid interactions. This binding mode accommodates variations in the NA catalytic pocket, contributing to the breadth of virus inhibitory activity and prophylactic protection against influenza A and B viruses in mice.

## Results

### Identification and characterization of mAbs from NA-specific B cells of a convalescent individual

The functionally and structurally conserved catalytic pocket of NA (Fig. 1a) and its relatively conserved antigenic properties makes it an attractive target for broadly cross-reactive antibodies. To isolate broadly cross-reactive B cells, we obtained cryopreserved peripheral blood mononuclear cells (PBMCs) from an individual, "donor A," who had a documented H3N2 infection prior to the 2015 sample collection time points. We identified NA-specific memory B cells by staining with the N4 NA of A/Red Knot/Delaware Bay/310/2016 (H10N4) and the N5 NA of A/shorebird/Delaware Bay/309/2016 (DB16, H10N5) of non-circulating avian influenza viruses, sorted single B cells by flow cytometry, and sequenced the corresponding B cell receptor heavy and light chain variable genes (Supplementary Figs. 1 and 2). Six antibodies were selected for further analysis, all originating from the same clonal family representing the largest expanded clone at both time points 1 and 2 (roughly one and 2 months after infection, respectively). This clonal family, termed NCS.1.x, had reactivity to non-circulating NA subtypes N4 and/or N5 and accounted for >90% of B cells isolated at time point 1 and >50% at time point 2 (Fig. 1b).

When recombinantly expressed as human immunoglobulin G1 (IgG1), all NCS.1.x mAbs exhibited broad binding to all N1 subtype NAs tested, including the sNAps[18] of A/New Caledonia/20/1999 (NC99, H1N1), A/California/04/2009 (CA09, H1N1), A/Michigan/45/2015 (MI15, H1N1), A/Vietnam/1203/2004 (VN04, H5N1), as well as an unmodified NA of VN04 (Fig. 1c). Notably, while limited, NCS.1.2 and NCS.1.3 displayed cross-reactivity to an N2 subtype NA of A/Wisconsin/67/2005 (WI05, H3N2). Additionally, all NCS.1.x mAbs demonstrated reactivity to the N5 DB16 NA which we used as one of the B cell probes, as well as the NA of the influenza B-Victoria lineage virus B/Colorado/06/2017 (B/Vic CO17). None of the NCS.1.x mAbs exhibited binding to the sNAp of A/Jiangxi-Donghu/346-2/2013 (H10N8, JD13) or the NA of A/Darwin/106/2021 (H3N2, DW21). The binding breadth of NCS.1.x mAbs appeared limited for the group 2 NA subtypes relative to 1G01[15], although the latter did not show appreciable binding to N5 DB16 nor N1 NC99.

Further characterization by biolayer interferometry (BLI) confirmed avid binding to N1-CA09-sNAp, N5 DB16, and B/Vic CO17 (Fig. 1d). None of the NCS.1.x mAbs tested bound to the unmodified N1 CA09 (Fig. 1d), likely due to its conformational instability[18]. However, robust binding by NCS.1.1 and NCS.1.3 was observed to the sNAp of the same strain, underscoring the importance of conformational stability in preserving the antigenicity of NA tetramers.

NCS.1.x antibodies use *IGHV4-31*03* and *IGKV2-28*01* genes for their Ig heavy and light chain variable genes, respectively. Phylogenetic tree analysis showed that the members of this clonal family exhibit varying degrees of maturation (Supplementary Fig. 2). Somatic hypermutations (SHM) on the nucleotide level revealed percent divergence from the germline ranging from 10.4% to 12.4% and 5.0% to 5.7% for the NCS.1.x heavy chain and light chain genes, respectively. Alignment of the Ig heavy chain amino acid sequences to the germline gene *IGHV4-31*03*-encoded sequence highlighted differences of 19, 18, and 18 positions for NCS.1, NCS.1.1, and NCS.1.3, respectively, with many shared substitutions (Fig. 1e). Of note, all characterized NCS.1.x mAbs shared identical CDRH3 sequences. Subsequent negative-stain electron microscopy (nsEM) analysis of NCS.1.1 in complex with N1-CA09-sNAp suggested that NCS.1.1 binds on the NA apical face, proximal to the highly conserved catalytic site (Fig. 1f). In summary, we isolated and characterized six mAbs from an individual with a recent H3N2 infection that reacted with N4/N5 NA probes and showed cross-reactivity with N1 and B-Victoria NAs, as well as limited reactivity against N2 NAs.

### NCS.1.1 recognizes the catalytic pocket of N1 NA via substrate mimicry

To elucidate the molecular basis for broad NA recognition by NCS.1.x mAbs, we used cryoEM to determine the structure of NCS.1.1 in complex with an N1 sNAp, N1-CA09-sNAp. Examination of 2D class averages revealed four NCS.1.1 Fabs bound to each NA tetramer for all observed N1 sNAp particles (Fig. 2a). The final 2.29 Å structure of this complex

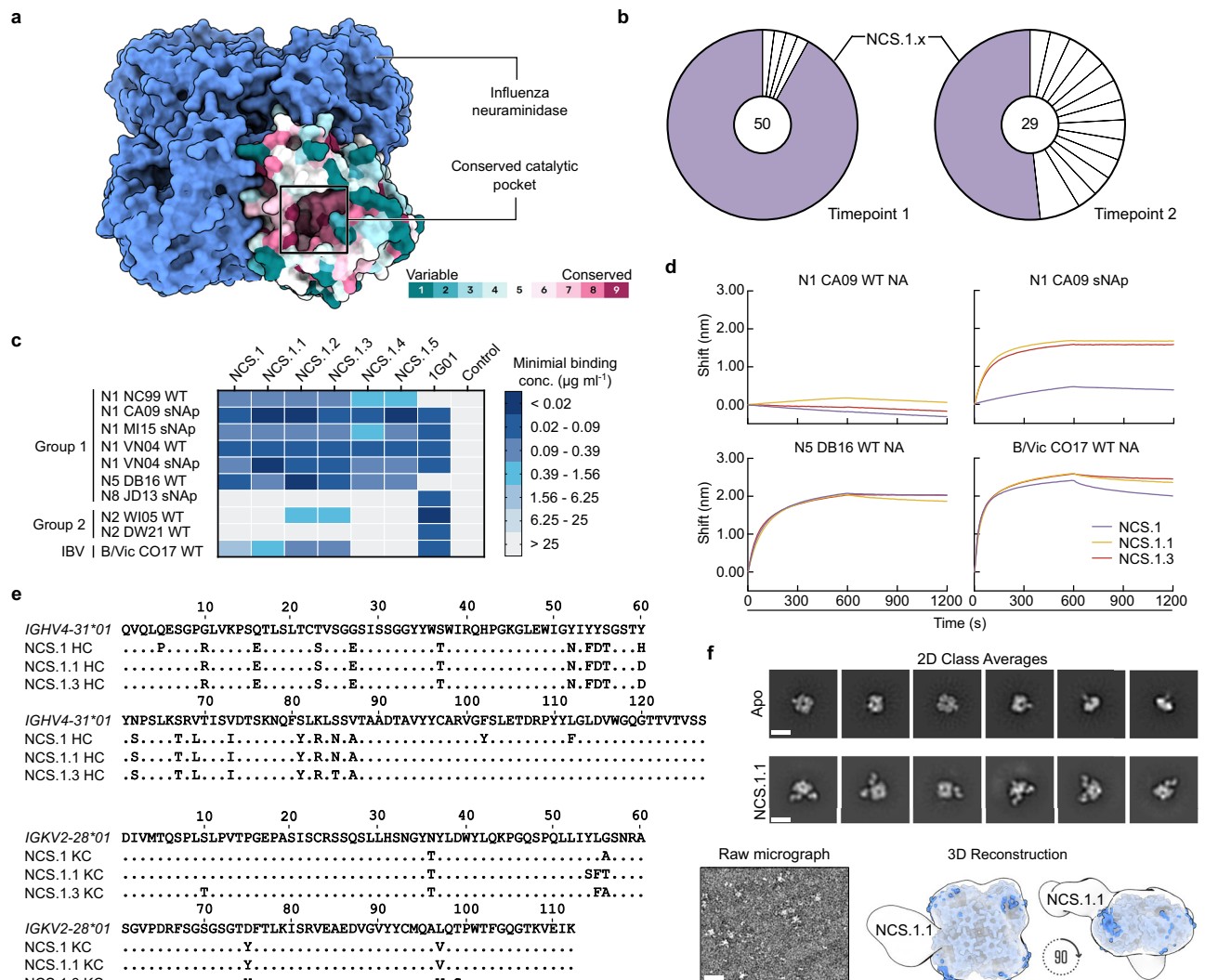

**Fig. 1 | Characterization of NA catalytic site targeting antibodies. a** Schematic model of influenza NA with its surface conservation across a single protomer. **b** Expanded B cell clones among sorted N4/N5-specific B cells. Each pie slice indicates a B cell clone with the same $V_H$ and $V_K/V_L$ gene usage and similar CDRH3 sequence. The total number of paired heavy- and light-chain sequences analyzed is shown inside each pie chart. Light purple pie slices indicate the expanded NCS.1.x clone. **c** Heat map of mAb binding to recombinant influenza A NAs of group 1, group 2, and influenza B NAs by ELISA. WT and sNAp refer to wild-type and stabilized NA proteins, respectively. D25[67] (anti-respiratory syncytial virus

mAb) was used as a negative control. **d** NA-binding of NCS.1.x mAbs measured by BLI. Binding was measured with recombinant NA and purified IgG. **e** Sequence alignment of immunoglobulin light- and heavy chain of NCS.1.x to their germline sequence (Kabat numbering). **f** 2D class averages of N1 CA09 sNAp alone (apo) and in complex with NCS.1.1 (top, scale bar: 80 Å). Representative region from a raw micrograph of NCS.1.1 bound to N1 CA09 sNAp (bottom left, scale bar: 240 Å). nsEM 3D reconstruction of the NCS.1.1–N1 CA09 sNAp complex (bottom right). Experiments were performed twice with similar results and results displayed from a representative experiment. Source data are provided as a Source Data file.

confirmed the presence of bound $Ca^{2+}$ and enabled the modeling of an intricate $H_2O$ network established along the epitope between N1 and NCS.1.1 (Fig. 2b, c, Supplementary Fig. 3, Supplementary Table 1). To ensure accurate water placement, we carried out a systematic validation pipeline (see Methods), confirming that all modeled waters were supported by clear density in both C4 and C1 symmetry reconstructions of the complex (Supplementary Fig. 3). NCS.1.1 predominantly engaged the N1 catalytic pocket via its CDRH3, with the interaction partially facilitated by coordinating water molecules. These molecules both stabilize the functional conformation of CDRH3 and mediate its interaction with the target epitope (Fig. 2b–d).

To contextualize NCS.1.x mAbs relative to other known NA catalytic-pocket binding mAbs, we conducted a comparative analysis of NCS.1.1 against mAbs FNI9[17] and 1G01[15]. A global comparison of binding footprints revealed that NCS.1.1 exhibits a distinctive binding footprint compared to FNI9, with the heavy and light chain

orientations swapped, resulting in an ~180° difference in binding angle (Fig. 2e, f). While NCS.1.1 and 1G01 demonstrate a remarkably similar approach angle (Fig. 2g), they diverge substantially in their CDRH3 sequences (Fig. 2h). By contrast, NCS.1.1 shares a higher level of sequence similarity in the CDRH3 region with FNI9, with both mAbs having structural and molecular features at the CDRH3 tip that promote substrate mimicry. Specifically, NCS.1.1 and FNI9 each incorporate Arg and Asp residues in their CDRH3 loops, enabling them to embed into the catalytic pocket (Fig. 2h). Despite this similarity, the residue order differs: NCS.1.1 features an Asp-Arg ($D100B_{HC}-R100C_{HC}$) motif, while FNI9 presents an Arg-Asp motif ($R106_{HC}-D107_{HC}$). However, through structural convergence, both achieve the same spatial configuration within the catalytic site (Fig. 2h), whereas 1G01 lacks this motif and thus engages with NA in a different manner (Fig. 2h).

In NCS.1.1, residue $D100B_{HC}$ within the Asp-Arg motif forms an ionic hydrogen bond network with $R118_{NA}$, $R293_{NA}$, and $R368_{NA}$ in the

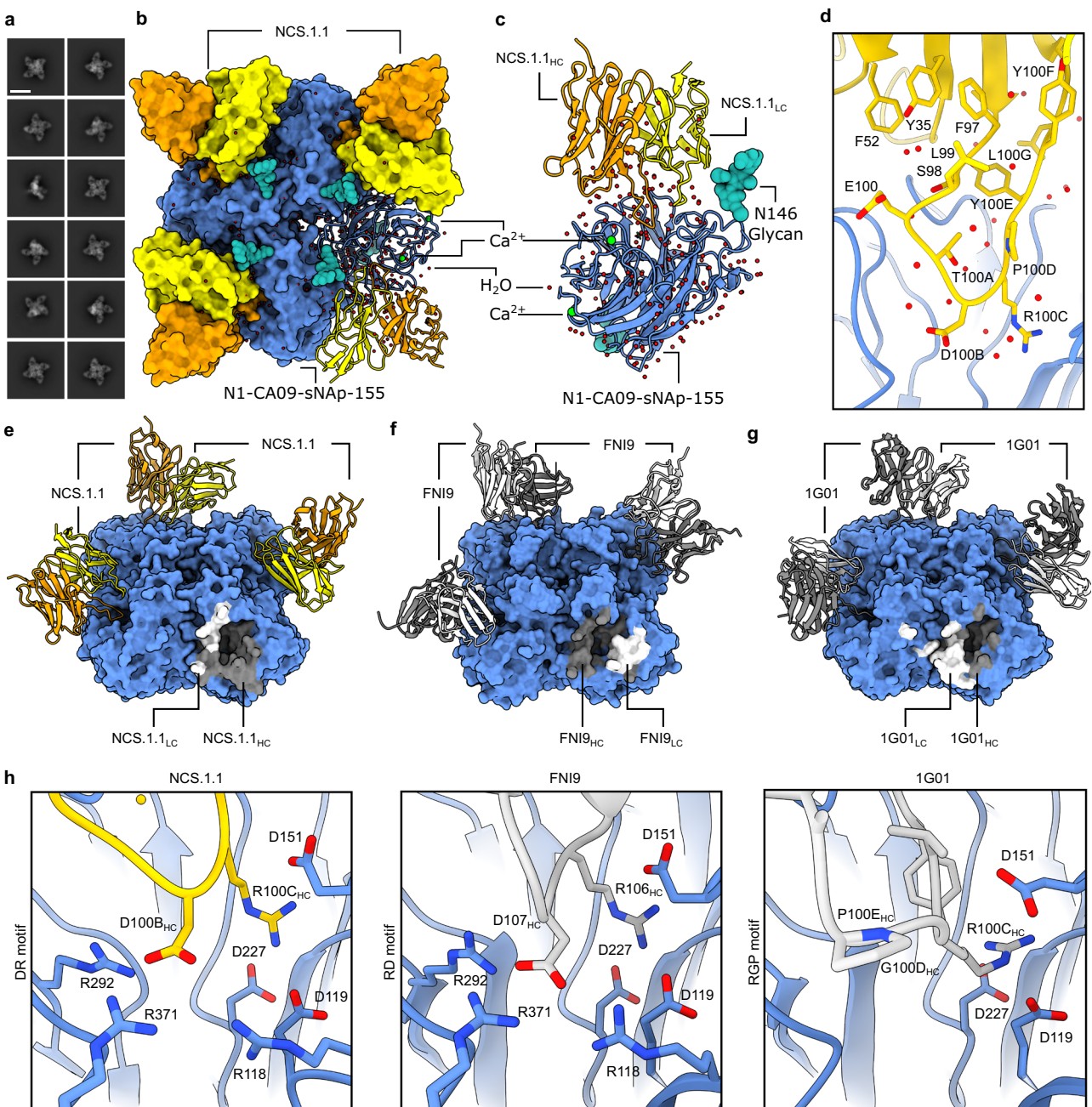

**Fig. 2 | cryoEM analysis of NCS.1.1 recognition of the NA catalytic pocket. a** 2D class averages of NCS.1.1 bound to N1-CA09-sNAp-155 showing four fabs bound to all particles. Scale bar, 180 Å. **b** 2.29 Å cryoEM reconstruction depicting the binding of NA-CA09-sNAp-155 to four copies of NCS.1.1. **c** Ribbon diagram illustrating the interaction of NCS.1.1 with a single protomer of N1-CA09-sNAp-155, with highlighted Ca²⁺, glycans, and H₂O molecules. **d** Close-up view of NCS.1.1 CDRH3 with water molecules (red spheres). **e**–**g** Comparative analysis of the binding footprints of NCS.1.1 (**e**), FNI9 (PDB: 8G3P) (**f**), and 1G01 (PDB: 6Q23)(**g**). **h** Comparison of CDRH3 binding motifs for embedding into the NA catalytic pocket for NCS.1.1, FNI9, and 1G01.

catalytic pocket, mimicking the sialic acid substrate's carboxyl (C1) group or the oseltamivir (OSE) carboxylate oxygen and hydroxyl groups (Fig. 3a). This network is similarly reproduced by D107$_{HC}$ in FNI9, highlighting the structural convergence used to coordinate Arg residues in the catalytic pocket between unrelated antibodies and the sialic acid substrate. Additionally, NCS.1.1 mimics sialic acid through the coordination of two H₂O molecules positioned between D277$_{NA}$ and T100A$_{HC}$ of the NCS.1.1 CDRH3 (Fig. 3b). This coordination of water molecules closely replicates interactions formed by the secondary alcohol groups at C8 and C9 of the glycerol-like chain in sialic acid. The engagement of H₂O with D277$_{NA}$ stabilizes the NA in a conformation identical to sialic acid bound state and is not a feature observed in NA structures in complex with either OSE or FNI9[17,19] (Fig. 3b). In OSE, two methyl groups occupy the same spatial coordinates resulting in shape complementarity rather than mimicry, while in FNI9, this space remains unoccupied, with a glycine at the same position as CDRH3 T100A$_{HC}$ of NCS.1.1. (Fig. 3b). Indeed, E277$_{NA}$ in the NCS.1.1 complex closely resembles its conformation when bound to the native substrate, whereas it adopts a different configuration in both FNI9- and OSE-bound structures[17,19].

Another feature of NCS.1.1 is that its light chain also appears to support the binding of CDRH3 to the catalytic pocket. The interaction

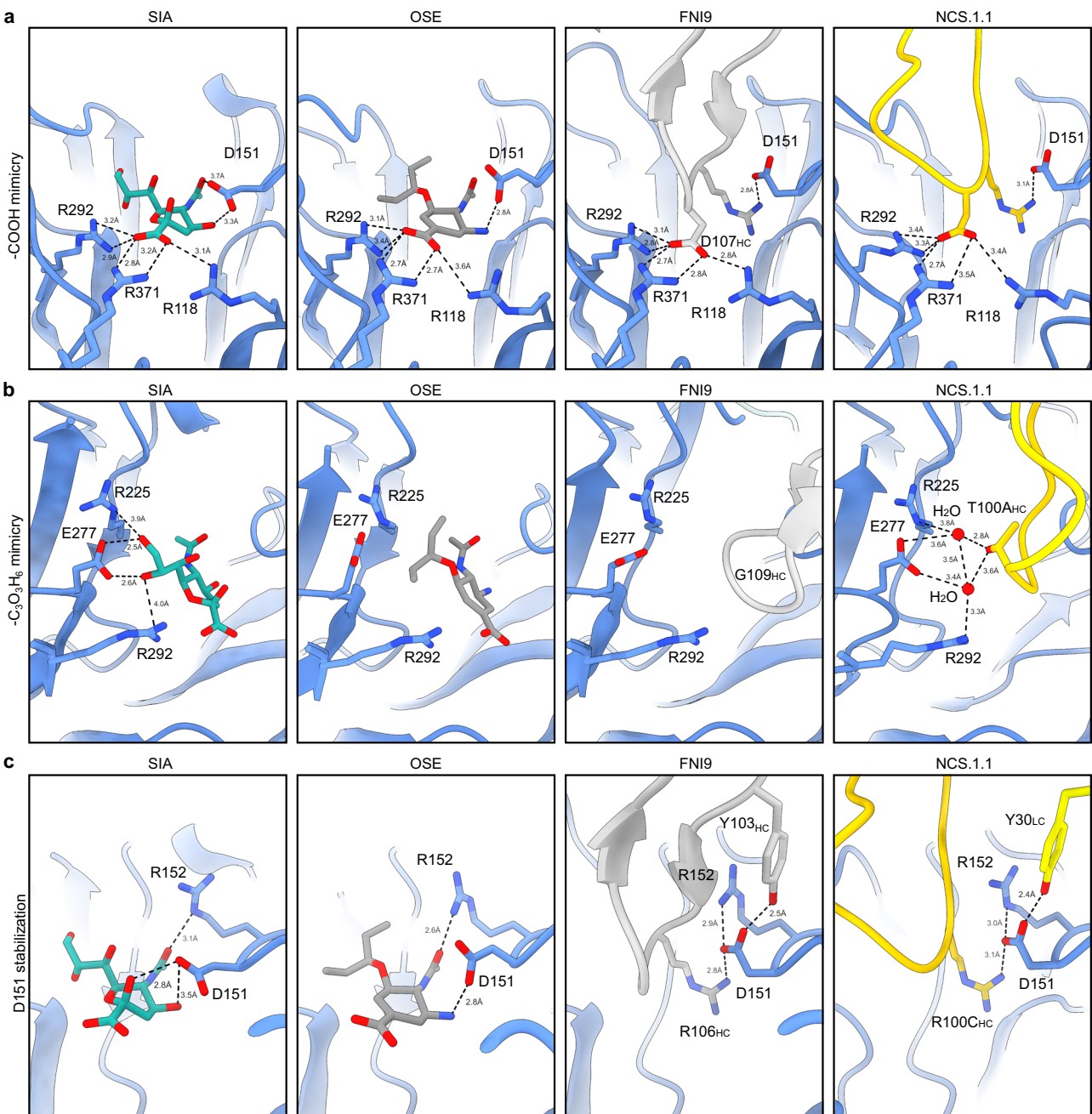

**Fig. 3 | Sialic acid receptor mimicry by NCS.1.1 is facilitated by a convergent evolutionary strategy. a–c** Comparative examination of the binding modes of sialic acid (SIA), Oseltamivir (OSE), antibodies FNI9, and NCS.1.1. Close up views of regions surrounding $R118_{NA}$, $R292_{NA}$, and $R371_{NA}$ (**a**), $E277_{NA}$ (**b**), and $D151_{NA}$ and $R152_{NA}$ (**c**).

between $Y30_{LC}$ on CDRL1 interacts directly with $D151_{NA}$ in a manner distinct from its conformation when bound by either sialic acid or OSE (Fig. 3c). This CDRL1 interaction mirrors findings observed in FNI9 (Fig. 3c). However, unlike NCS.1.1, where CDRL1 is responsible for this interaction, FNI9 employs $Y103_{HC}$ within its CDRH3 to help coordinate $D151_{NA}$. In both mAbs, stabilizing $D151_{NA}$ in this alternative conformation helps promote interaction with the CDRH3 Arg ($R100C_{HC}$ in NCS.1.1 or $R106_{HC}$ in FNI9) within its Asp-Arg/Arg-Asp motif (Fig. 3c). The stabilization of $D151_{NA}$ is reinforced through coordination with $R152_{NA}$. In the FNI9-bound structure, the rotameric conformation of $R152_{NA}$ aligns closely with the OSE-bound NA, while in the NCS.1.1-bound structure, it more accurately reflects the sialic acid-bound conformation, achieving a greater degree of structural mimicry within NA (Fig. 3c). This mimicry, unique to NCS.1.x, distinguishes it from both OSE- and FNI9-bound NA structures[17,19].

## Water-assisted lock-and-key binding enables NCS.1.x-like mAbs to recognize NA variants

To explore how NCS.1.x mAbs recognize other subtype NA subtypes, we performed cryoEM analysis of NCS.1 in complex with an N5 NA (DB16), one of the probes used during initial B cell sorting. cryoEM 2D class averages and preliminary 3D reconstructions showed that NCS.1 Fabs formed stable complexes with recombinant N5 NA, with most N5 NA molecules bound by four Fabs and a subset bound by three (Fig. 4a, b, Supplementary Fig. 4). A resulting 3.36 Å cryoEM structure of the fully occupied NA-Fab complex revealed that the binding of NCS.1 to the N5 catalytic pocket closely resembles that of NCS.1.1 to N1, with similarities not only in the targeted epitope but also in the approach angle of the Fab to the catalytic site (Fig. 4c, d, Supplementary Table 1). This similarity in epitope recognition and binding orientation suggests a largely conserved mode of interaction within the NCS.1 family,

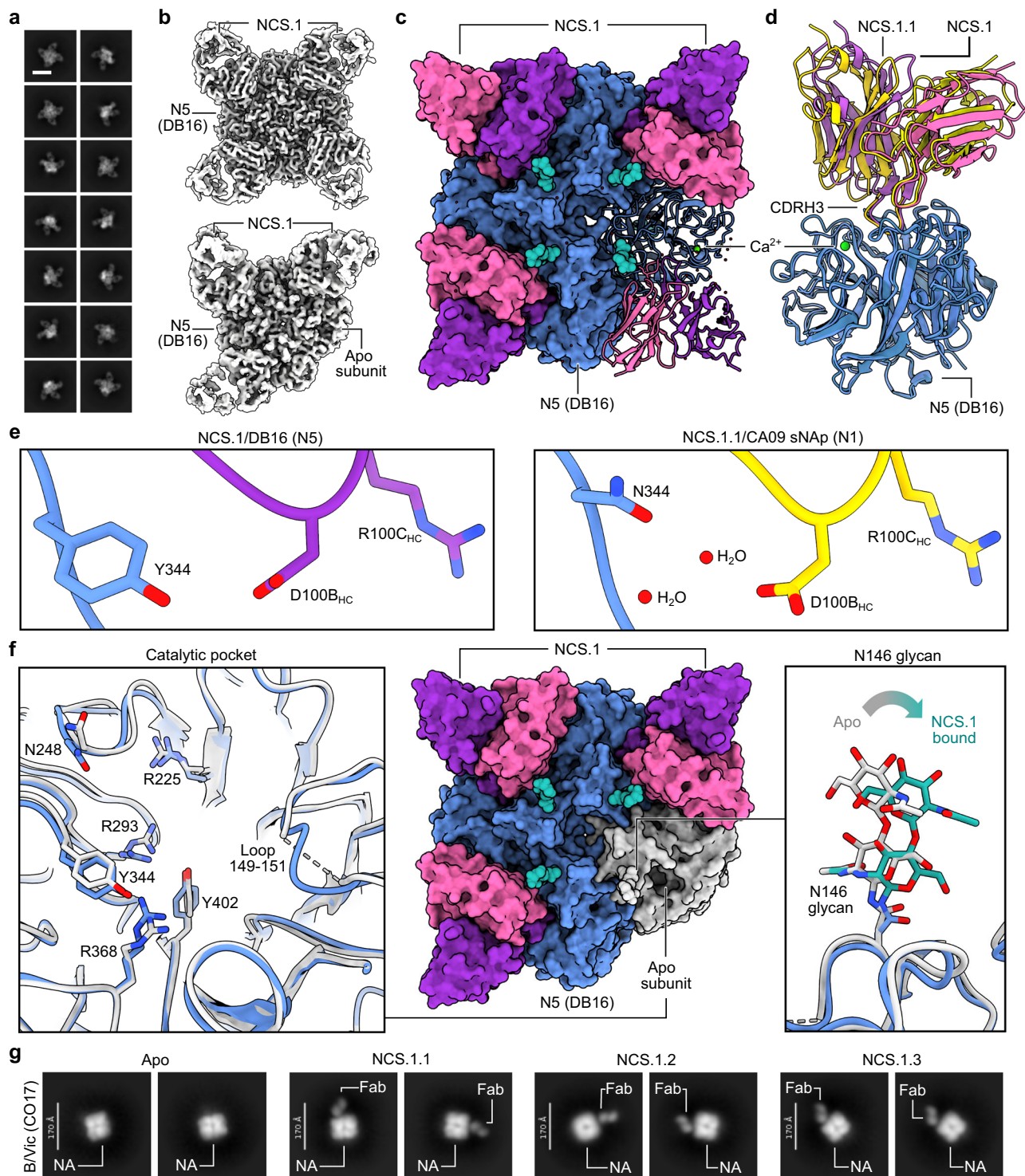

**Fig. 4 | Water-mediated adaptations in NCS.1-like mAb recognition enable broad targeting of structurally conserved NA epitopes across group 1 antigens. a** 2D class averages of NCS.1 bound to WT N5 DB16. Scale bar, 180 Å. **b** CryoEM 3D reconstructions showing two populations in the dataset: one with all four binding sites occupied, and one with three binding sites occupied. **c** 3.36 Å cryoEM reconstruction illustrating the binding of WT N5 DB16 to four copies of NCS.1. **d** Ribbon diagram highlighting similarities in binding between NCS.1.1/N1 and NCS.1/N5. **e** Structural comparison of NCS.1/N5 and NCS.1.1/N1. **f** Analysis of the 4.1 Å three-Fab bound map of NCS.1 bound to N5. **g** nsEM 2D class averages of apo, NCS.1.1-, NCS.1.2-, and NCS.1.3-bound NA of B/Vic CO17. Scale bars, 170 Å.

optimized for engaging the catalytic pocket across NA subtypes (Fig. 4d). Comparative analysis of N1- and N5-bound structures reveals that NCS.1.x mAbs achieve breadth partly through water-mediated interactions that adapt to subtype-specific residue variations in the catalytic pocket. Specifically, position 344 differs: DB16/N5 has a bulky Tyr, while CA09/N1 has a smaller Asn. The absence of Y344 in CA09/N1

leaves a void which is effectively filled by two ordered $H_2O$ molecules (Fig. 4e).

The 4.18 Å structure of the N5 protomer bound by three Fabs provided further insight into binding characteristics, revealing that NCS.1 binding induces minimal structural changes in the catalytic pocket, supporting a lock-and-key binding mode (Fig. 4f,

Supplementary Fig. 4, Supplementary Table 1). Minor shifts in adjacent residues and consistent glycosylation patterns, with slight positional adjustments in glycan N146, allowed for precise binding without major conformational adjustments (Fig. 4f). Finally, negative-stain EM analysis of NCS.1.x mAbs revealed that the structural recognition of the NA catalytic pocket is conserved across influenza B NA, consistent with ELISA binding data (Figs. 1c, 4g). These findings underscore that the cross-reactivity of NCS.1.x mAbs towards group 1 NA subtypes is achieved through a water-assisted, substrate-mimicking lock-and-key binding that precisely targets the conserved catalytic pocket.

## In vitro functionality of NCS.1, NCS.1.1, and NCS.1.3

We next assessed the functional capabilities of the NCS.1.x mAbs against various influenza viruses. Initially, we evaluated the NA inhibitory (NAI) activity by using the influenza replication inhibition NA-based assay

(IRINA)[20]. This assay detects the enzymatic activity of NA present on the surface of virus-infected cells, providing a direct functional measurement through the cleavage of 2′-(4-methylumbelliferyl)-α-D-N-acetylneuraminic acid (MUNANA) to the fluorescent 4-methylumbelliferone (4-MU). IRINA was performed with multiple influenza A viruses carrying group 1 subtype NAs and two influenza B viruses from the Victoria lineage. All NCS.1.x mAbs tested exhibited NAI activity against H1N1 and H5N1 viruses in this assay comparable to the control 1G01, which also targets the catalytic pocket (Fig. 5a). The G248R mutation of A/New York/146/2000 H1N1 did not impact binding of the NCS.1.x mAbs. This amino acid change is centrally located within the heavy chain footprint of the NCS.1.x mAbs (Supplementary Fig. 5a). Structural analysis reveals that despite its central location, G248 mutations are accommodated by conformational flexibility within the CDRH1 loop of the antibody. The glycine-serine–rich motif of CDRH1 facilitates structural adaptation,

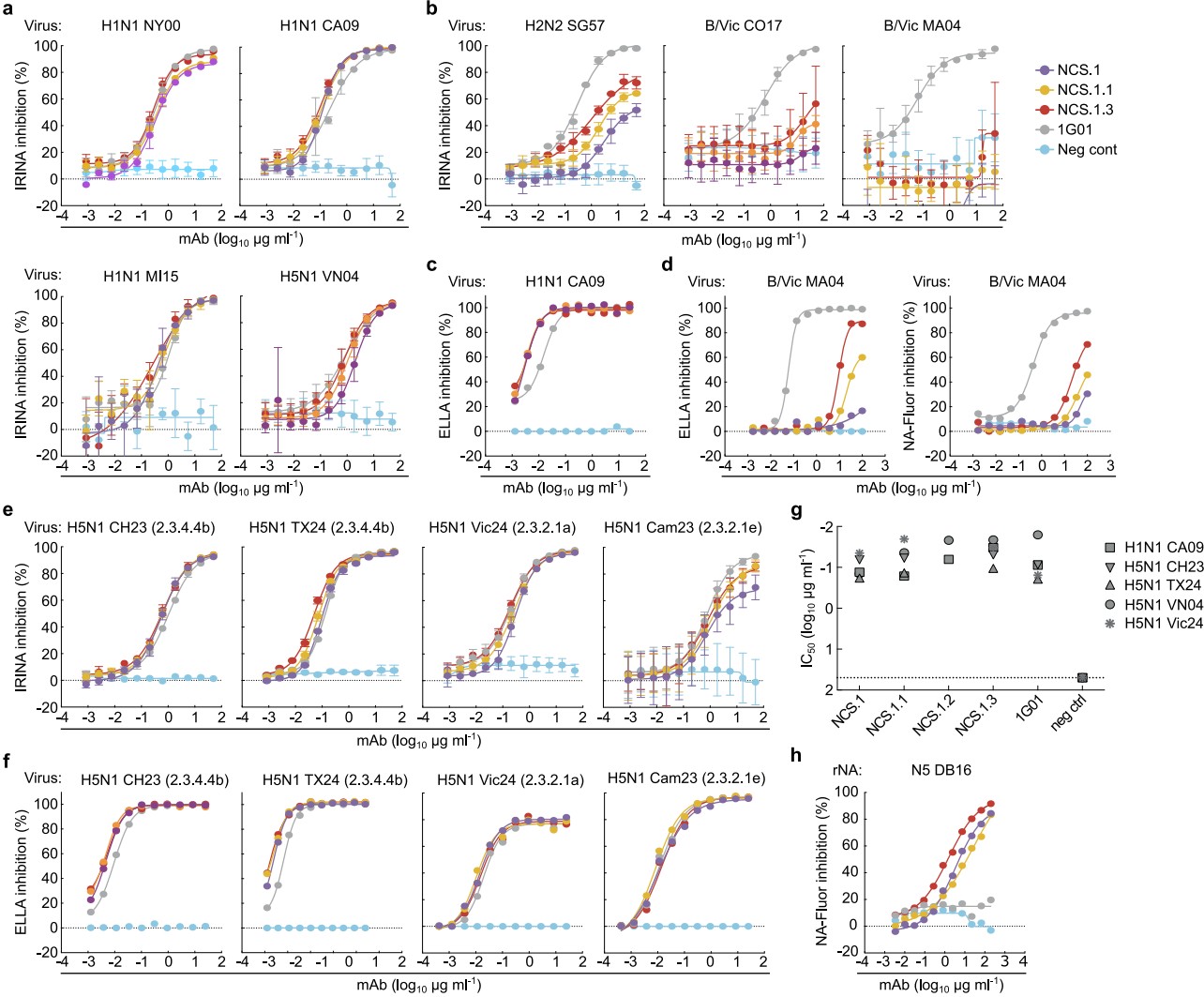

**Fig. 5 | In vitro characterization of NCS mAbs. a** NAI activity of the mAbs against viruses encompassing N1 NA measured by IRINA. H1N1 NY00, H1N1 CA09, H1N1 MI15 and H5N1 VN04 reporter influenza viruses were used. Data are plotted as mean ± SD of $n = 4$ individual wells at each dilution. **b** NAI activity of the mAbs against H2N2 and influenza B viruses. H2N2 SG57, B/Vic CO17, B/Vic MA04 reporter influenza viruses were used. Data are plotted as mean ± SD of $n = 4$ individual wells at each dilution. **c** NAI activity of the mAbs measured by ELLA with H1N1 CA09 reporter influenza virus. Data are plotted as mean of $n = 2$ individual wells at each dilution. **d** NAI activity of the mAbs measured by ELLA and NA-Fluor assay with B/Vic MA04 wild-type influenza virus. Data are plotted for each dilution. **e, f** NAI activity of the mAbs against clades 2.3.4.4b, 2.3.2.1a and 2.3.2.1e H5N1 reporter influenza viruses measured by IRINA (**e**) and ELLA (**f**). H5N1 CH23, H5N1 TX24, H5N1 Vic24 and H5N1 Cam23 reporter viruses were used. Data are plotted as mean ± SD of $n = 4$ individual wells (IRINA) and as mean of $n = 2$ (ELLA) at each dilution. **g** Viral growth inhibitory activity of the mAbs against H1N1 CA09, H5N1 VN04, H5N1 CH23, H5N1 TX24 and H5N1 Vic24 reporter influenza viruses. The half-maximal inhibitory concentration (IC$_{50}$) of mAb for each virus is shown. $n = 4$ individual wells at each dilution. **h** NAI activity of the mAbs measured by NA-Fluor assay with recombinant NA N5 DB16 protein. Data are plotted as mean of $n = 2$ individual wells at each dilution. All experiments were performed once including appropriate controls. Source data are provided as a Source Data file.

allowing the loop to avoid steric clashes introduced by bulkier side chains. In our N5–DB16 complex, which harbors a related G248S mutation, this rearrangement also displaces water molecules from the original binding site while preserving key interactions through alternative hydrogen bonding or van der Waals contacts (Supplementary Fig. 5b, c). However, IRINA inhibition activity of NCS.1.x mAbs was less potent for influenza B viruses compared to 1G01 (Fig. 5b). Given that IRINA measures direct competition of catalytic pocket, we also performed an enzyme-linked lectin assay (ELLA), which assesses the cleavage of sialic acid from the highly sialylated glycoprotein fetuin by NA[21]. This cleavage can be inhibited both directly or indirectly through steric hindrance by antibodies. NCS.1.x mAbs demonstrated ELLA inhibition activity similar to 1G01 against H1N1 CA09 (Fig. 5c). However, despite detecting ELISA binding, weak ELLA inhibition activity against influenza B viruses CO17 or B/Malaysia/2506/2004 (MA04) by NCS.1.x mAbs was observed only at higher mAb concentrations (Fig. 5d).

Given the ongoing global panzootic H5N1 outbreak, we next assessed the potential utility of the NCS.1.x mAbs against clades 2.3.4.4b, 2.3.2.1a and 2.3.2.1e H5N1 viruses. Both IRINA and ELLA were performed with A/Chile/25945/2023 (2.3.4.4b H5N1, CH23), A/Texas/37/2024 (2.3.4.4b H5N1, TX24), A/Victoria/149/2024 (2.3.2.1a H5N1, Vic24) and A/Cambodia/NPH230032/2023 (2.3.2.1e H5N1, Cam23) reporter viruses which were engineered using the HA and NA sequences of isolates from recent human H5N1 cases in Chile[22], Texas[23], Victoria[24], and Cambodia[25], respectively. All NCS.1.x mAbs as well as 1G01 were highly potent against H5N1 viruses, representing three different clades both in IRINA and ELLA (Fig. 5e, f). Other catalytic site-targeting antibodies, including FNI9, 1G05[26], 2E01[26], and DA03E17[16], were also tested and effectively inhibited NA catalytic activity against various N1 subtypes, including clade 2.3.4.4b H5N1 CH23 and TX24 (Supplementary Fig. 6). We then assessed whether NCS.1.x mAbs inhibit in vitro viral propagation of H1N1 CA09 and four H5N1 viruses, including A/Vietnam/1203/2004 (clade 1 H5N1, VN04), CH23, TX24, and Vic24. The virus growth inhibition assay utilized reporter influenza viruses and infected cells were monitored over time[27]. All NCS.1.x mAbs demonstrated inhibitory activity against all tested viruses with similar

potency as 1G01 (Fig. 5g). Importantly, the ability of these mAbs to effectively suppress clades 2.3.4.4b and 2.3.2.1a H5N1 viruses underscores their potential utility in developing effective countermeasures against emerging influenza viruses with pandemic potential. We additionally demonstrated that all NCS.1.x mAbs demonstrated ELLA inhibition activity against recombinant N5 DB16 (Fig. 5h), confirming our ELISA binding data (Fig. 1c).

## In vivo protective efficacy of NCS.1.x mAbs in mice

To evaluate whether the structural observations and in vitro inhibitory activity translated to in vivo protection, we next conducted a series of lethal influenza virus challenge studies in mice. Protective efficacy was assessed in a prophylactic challenge model where the mAbs (10 mg kg$^{-1}$) were administered intraperitoneally 24 h prior to intranasal infection with either A/California/07/2009 H1N1 (15 × 50% lethal dose ($LD_{50}$)), B/Malaysia/2506/2004 (>10 × $LD_{50}$), A/Vietnam/1203/2004 H5N1 (6 × $LD_{50}$), or A/dairy cattle/Texas/24008749001/2024 clade 2.3.4.4b H5N1 (3.5 × $LD_{50}$) (Fig. 6a). NCS.1, NCS.1.1, and NCS.1.3 conferred full protection against H1N1 CA09 infection, with <10% weight loss observed, comparable to the positive control (the anti-HA mAb MEDI8852[28]), whereas all mice receiving the negative control mAb succumbed to infection (Fig. 6b). In B MA04 challenge, both NCS.1.1 and NCS.1.3 provided complete protection with <10% weight loss (NCS.1.1) or no noticeable weight loss (NCS.1.3). In contrast, NCS.1 provided 80% (8 out of 10 mice) protection against the influenza B virus challenge, despite showing the largest weight loss among the NCS.1.x mAbs, consistent with its weak virus inhibitory activity against influenza B viruses in in vitro assays (Fig. 6c). In the H5N1 VN04 challenge, NCS.1.1 displayed the highest protective efficacy among all mAbs tested (70%; 7 out of 10 mice), surpassing the survival rates of NCS.1 (40%), NCS.1.3 (50%), and 1G01 (60%) (Fig. 6d). The positive control group which received an anti-HA stem bnAb had 90% survival, with a weight loss pattern similar to that of the NCS mAbs. Moreover, both NCS.1.1 and NCS.1.3 conferred partial protection against the A/dairy cattle/Texas/24008749001/2024 clade 2.3.4.4b H5N1 virus challenge, a viral strain derived from the recent outbreak in dairy cattle

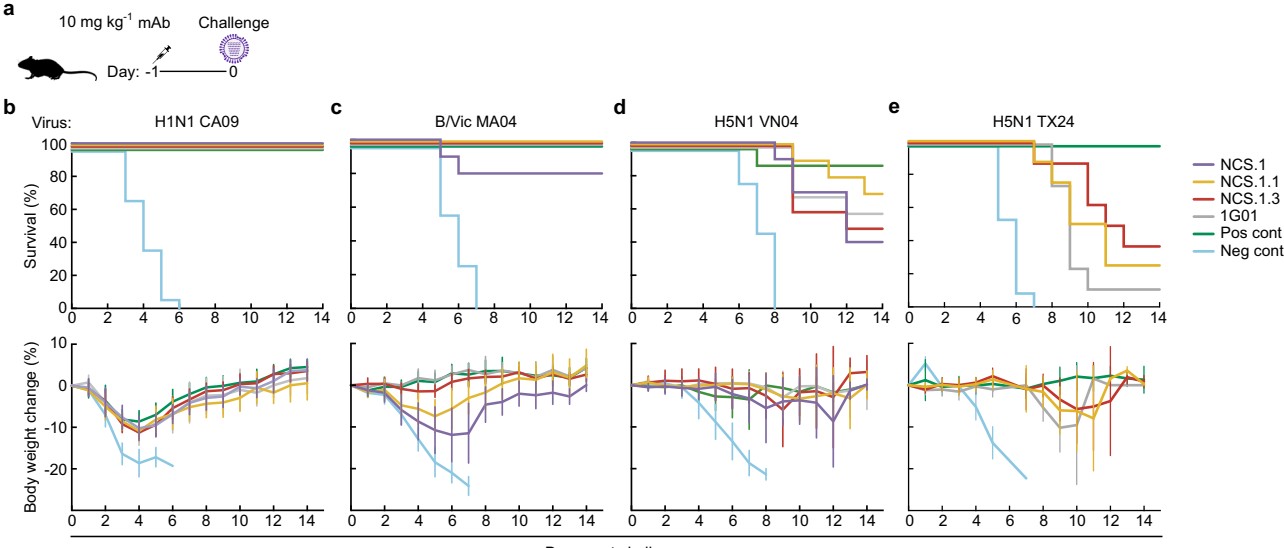

**Fig. 6 | In vivo protection by NCS.1, NCS.1.1 and NCS.1.3. a** Pre-exposure prophylaxis experiment in a murine model. **b–e** BALB/c mice (n = 8–10 per group) were administered 10 mg kg$^{-1}$ of mAbs intraperitoneally 24 h prior to intranasal viral challenge. Virus dose was 15 × $LD_{50}$ for H1N1 CA09 (**b**), >10 × $LD_{50}$ for B/Vic MA04 (**c**), 6 × $LD_{50}$ for H5N1 VN04 (**d**), and 3.5 × $LD_{50}$ for H5N1 TX24 (**e**). The upper panels show Kaplan–Meier survival curves, and the lower panels show percent body weight loss (data are plotted as mean ± SD). For the H1N1 CA09 and H5N1 VN04

challenges, VRC01 served as the negative control and 315-02-1H01[68] as the positive control. In the B/Vic MA04 challenge, PGT121[69] and mAb47 were used as negative and positive controls, respectively. For the H5N1 TX24 challenge, VRC01[70] and MEDI8852[28] were used as negative and positive controls, respectively. All challenge experiments were performed once. Statistical analyses of Kaplan–Meier curve comparisons are shown in Supplementary Table 2. Source data are provided as a Source Data file. Mouse and syringe icons were generated using BioRender.

 

across the U.S., showing slightly better protection on average as compared to 1G01, which provided only 10% protection (Fig. 6e). The negative control group had 0% survival, while the positive anti-HA stem bnAb control showed 100% survival, indicating that the bnAb MEDI8852 targeting the HA stem has superior protective efficacy against this virus, potentially due to its ability to block viral entry. Notably, all groups treated with NA mAbs exhibited extended time to euthanasia compared to the negative control, with delayed onset of weight loss or neurological symptoms by ~4 days, with NCS.1.3 demonstrating the greatest effect. This H5N1 2.3.4.4b virus is reported to cause neurological symptoms meeting euthanasia criteria[29,30], highlighting the importance of blocking the viral infection to prevent the development of neurological symptoms in small animals. Overall, when administered prophylactically, NCS.1.x mAbs provided broad protection across influenza A and B viruses, with partial protection against H5N1 strains, including the highly neurovirulent clade 2.3.4.4b virus.

## Discussion

In this study, we isolated a set of mAbs that target the NA catalytic site through precise molecular mimicry of its native sialic acid substrate and evaluated their function both in vitro and in vivo. These mAbs demonstrated robust prophylactic protection in murine models, providing insights that could lead to the development of effective tools for combating influenza infection.

Near-atomic resolution cryoEM analysis revealed that NCS.1.x mAbs bind the conserved NA catalytic site through precise sialic acid mimicry. Both NCS.1.x and the recently identified FNI9[17] mAbs, despite differences in origin, immunogenetics, and structure, share a conserved Asp-Arg/Arg-Asp motif within the CDR3 loop, underscoring its essential role in enabling molecular mimicry of sialic acid. The accompanying manuscript by Jo et al. further elaborates this structural motif in additional antibodies—such as 1G05[26], Z2B3[31], and DA03E17—highlighting its recurrence across diverse antibody families targeting the NA catalytic pocket and underscoring the implications of this prevalence for future vaccine design. In addition to this motif, our analysis further identified a structurally conserved Tyr located in the CDRH3 loop of FNI9[17] and in the CDRL1 of NCS.1.x mAbs. These shared structural features spanning multiple antibody families illustrate the convergent pathways to achieve broad specificity and offer valuable insights for future antibody discovery and development.

Beyond conserved motifs, water molecules seem crucial for stabilizing interactions between the CDRH3 of NCS.1.x mAbs along the binding interface. Our data suggest that, NCS.1.x mAbs incorporate water molecules to stabilize the CDRH3 loop and form bridging contacts along the CDRH3 interface within the catalytic pocket. While the CDRH3 loop is essential for effective binding to this buried epitope, its structural flexibility is somewhat limited by constraints of the protein backbone and available amino acid repertoire. Water molecules help overcome these limitations by bridging structural gaps with hydrogen bonds, enabling closer interaction with catalytic pocket residues than what amino acids alone could achieve[32,33]. Importantly, water-mediated interactions along binding interfaces are well known to enhance binding affinity, which can ultimately contribute to the robustness of the immune response[34,35]. As a result, we anticipate that water molecules may be more commonly incorporated along binding interfaces in NA catalytic site antibodies than current reports suggest. This is likely influenced by resolution limitations in existing structural studies and by atomic modeling choices where water molecules were not included. As more high-resolution structures with modeled water molecules become available, deeper insights into water's role in stabilizing antibody-antigen interactions and broadening antibody specificity could help drive innovations in therapeutic design, antibody engineering, and protein design strategies that account for solvent interactions.

The high resolution of our structure of NCS.1.1 in complex with N1-CA09-sNAp enabled us to visualize many bound waters and enabled us to better understand their role in antigen recognition by NCS.1.x antibodies. Specifically, the incorporation of water molecules along the CDRH3 interfaces of NA catalytic pocket-targeting antibodies may help broaden specificity by compensating for variations in NA catalytic pocket residues. This adaptability could be achieved by adding or removing specific water-mediated contacts, as demonstrated by NCS.1.x mAbs reported in this study. Indeed, broadly neutralizing antibodies against HIV-1 (e.g., VRC01-class and PG9) and influenza HA (e.g., C05) utilize water molecules to bridge interactions with structurally diverse antigens, allowing them to accommodate antigenic variation while maintaining high-affinity binding[36–39]. Future high-resolution cryoEM studies of NCS.1.x mAbs bound to additional NA subtypes, including B/Vic CO17, could further clarify the role of water-mediated interactions in broad NA targeting. Similarly, studies of other catalytic site-targeting mAbs may offer additional insights if water molecules are explicitly modeled. While our structural data highlight the role of water molecules in bridging contacts along the binding interface, we recognize that they are likely not the sole determinant of breadth. Other factors, such as CDR loop flexibility, length, and its amino acid composition, and the conserved interactions through the CDRH3 motif with the catalytic site, likely also play major roles[40,41]. As more detailed high-resolution structures become available, gaining deeper insights into water's role in stabilizing antibody-antigen interactions and expanding antibody breadth could help drive innovations in therapeutic design.

Beyond structural insights, the functional breadth of NCS.1.x mAbs also warrants consideration. Our data demonstrate that NCS.1.x mAbs isolated following H3N2 infection exhibit cross-reactivity with group 1 NAs, suggesting therapeutic potential beyond group 1. We also observed some reactivity against both N2 and B strains, aligning with recent findings for FNI9 and DA03E17[16,17]. Although the precise mechanisms underlying the NCS.1.x-encoding B cell expansion after H3N2 infection remain unclear, it is noteworthy that the original samples were collected from donor A in 2015. This timing leads us to speculate that certain H3N2 strains circulating around that time may have binding affinity to NCS.1.x memory B cells, prompting robust clonal expansion during an initial response to infection. Future studies will be needed to gain better understanding of the developmental pathways for NA-specific B cell responses upon influenza infection.

This study also highlights the value of sNAps in assessment of antibody cross-reactivity and structure, especially where wild-type recombinant NAs have traditionally been limited by structural instability[18]. sNAps enabled us to identify NCS.1.x mAb reactivity with specific NA strains—particularly the CA09 strain used for high-resolution cryoEM analysis—that would have been challenging to characterize using wild-type NAs alone. This underscores the importance of sNAps as a reliable platform for probing interactions between NA and NA-targeting antibodies, which we expect will be important for future antibody discovery and structural studies[18,42].

Influenza outbreaks, particularly those involving viruses with high mortality, morbidity, and transmissibility, continue to pose significant global health risks due to their potential for rapid spread and severe outcomes. Among these, avian H5N1 influenza viruses are especially concerning, given their high case fatality rate and capacity for inter-species transmission among mammalian species. The potential of NCS.1.x and other NA catalytic site targeting mAbs to provide protection against a panzootic H5N1 outbreak is promising; however, further research is needed to account for species-specific differences in viral pathogenesis[3,5,43–45]. While mouse models of 2.3.4.4b H5N1 infection show high viral replication with limited systemic spread, ferrets and non-human primates exhibit variable susceptibility, and calves tend to experience more severe disease than pigs[3,30]. Most human cases with the "bovine" H5N1 virus have so far resulted in

conjunctivitis and mild overall disease[46–48]. These differences emphasize the need for diverse models in assessing promising medical countermeasures and continued effort to develop animal models reflective of human diseases. Lessons from the COVID-19 pandemic highlight the potential advantages of targeting multiple viral epitopes over a single site or antigen. Simultaneous targeting of both NA and HA by mAbs or vaccines would be advantageous to counter rapid viral evolution, and mitigate the risk of virus escape. Indeed, studies report synergistic effects when antibodies target multiple viral proteins together[17,49], and several anti-NA catalytic site antibodies—including NCS.1.x—have now shown a range of specificities and breadth[15–17,26,31].

Finally, antibodies targeting the NA catalytic site, such as NCS.1.x, FNI9, 1G01, and DA03E17, offer several advantages over traditional NA inhibitors. These antibody-specific benefits include longer half-life, targeted biodistribution, and reduced renal clearance[50,51] over the small molecule compounds, making them especially suited for prophylactic use in high-risk populations. As research advances, broadly reactive antibodies and vaccines that target the NA catalytic pocket, particularly when combined with other antiviral strategies, could become a cornerstone of pandemic preparedness, offering a versatile and powerful tool for both prevention and rapid response to emerging influenza threats.

## Methods

### Human specimens

The human PBMC sample used in this study was obtained under the study, VRC 200, a protocol for apheresis and specimen collection procedures to obtain plasma, PBMCs and other specimens for research studies (ClinicalTrials.gov identifier NCT00067054) at the National Institutes of Health (NIH) Clinical Center by the Vaccine Research Center (VRC) Clinical Trials Program, National Institute of Allergy and Infectious Disease (NIAID), NIH in Bethesda, MD. The protocol was reviewed and approved by the NIH Institutional Review Board. Informed consent was obtained from every enrolled participant and conduct of the study complied with all relevant ethical regulations. Compensation was provided to participants for their time and effort related to participation in this clinical trial research study. Blood samples of donor A (male, born in 1956) were collected in 2015 (interval between time point 1 and 2 was 4 weeks). Time point 1 sample was collected two to 6 weeks post-confirmed influenza positive PCR test. This donor PBMC sample was also used to isolate N2 NA-specific B cells in our previous study[42].

### Ethical approval for animal models

All experimental procedures conducted at University of Melbourne were approved by the University of Melbourne Animal Ethics Committee (protocol #22954). All animal procedures conducted at University of Pittsburgh were approved by the University of Pittsburgh Institutional Animal Care and Use Committee, IACUC (protocol #22040682), with animal care in accordance with the Guide for the Care and Use of Laboratory Animals (National Research Council) and the Association for Assessment and Accreditation of Laboratory Animal Care (AAALAC). All mice in this study were housed in AAALAC accredited animal facilities in a 12-h light/dark cycle at an ambient temperature of $22.2 \pm 2.8\,°C$ with a relative humidity maintained between 30–70%. Euthanasia was performed according to American Veterinary Medical Association guidelines, based on weight loss (>20% from baseline) and morbidity.

### Animal models and experimental procedures

At University of Melbourne, female BALB/c mice (8–10 weeks old) were used to examine the prophylactic potential of mAbs. The challenge virus, B/Malaysia/2506/2004, was grown in embryonated eggs, and virus titers were determined using 50% tissue culture infective dose ($TCID_{50}$) assays. Mice received intraperitoneal mAb injections at a dose

of 10 mg kg$^{-1}$. Twenty-four hours later, they were anesthetized using isoflurane and intranasally challenged with 50 µl of B/Malaysia/2506/2004 virus ($1 \times 10^6$ $TCID_{50}$) which is a predicted $LD_{50}$ between 10 and 100. Mice were monitored for weight loss and signs of infection for 14 days, and euthanized if they lost >20% of their pre-infection body weight. At University of Pittsburgh, female BALB/c mice (6–8 weeks old) were purchased from Jackson Laboratories (Bar Harbor, ME) and housed in standard microisolator caging in the ABSL-2 (H1N1 challenge) or ABSL-3 (H5N1 challenge). A/California/07/2009 (H1N1) was propagated in embryonated chicken eggs, aliquoted, and stored for all studies while infectious clones of A/Vietnam/1203/2004 and A/dairy cattle/Texas/24008749001/2024 (H5N1) were rescued and propagated in embryonated eggs prior to storage. While preliminary studies found no differences in survival or survival time between male and female mice, $LD_{50}$ doses of 5300 pfu for CA09 challenge and 2 pfu for H5N1 VN04 and TX24 challenges used in this study were established by using female mice, and hence only female mice were used in this study. Twenty-four hours prior to challenge, mAbs were intraperitoneally administered at a dose of 10 mg kg$^{-1}$. Virus challenges were delivered intranasally, with 50 µl applied to the nares while the mice were anesthetized by inhaled isoflurane. Back titrations confirmed that mice received $8 \times 10^4$ pfu ($15 \times LD_{50}$) of A/California/07/2009, 12 pfu ($6 \times LD_{50}$) of A/Vietnam/1203/2004, or 7 pfu ($3.5 \times LD_{50}$) of A/dairy cattle/Texas/24008749001/2024, respectively. Mice were euthanized if moribund, experiencing respiratory distress, or exhibiting >20% weight loss or hindlimb paralysis.

### Single-cell sorting, immunoglobulin amplification, sequencing and mAb production

Cryopreserved PBMC samples were thawed, stained with monoclonal antibodies to cell surface markers, and antigen-specific memory B cells were single-cell sorted as described in Lederhofer et al.[42]. The NA probe N4 (A/Red Knot/Delaware Bay/310/2016; N4 DB16) and N5 (A/Shorebird/Delaware Bay/309/2016; N5 DB16) were conjugated with either PE or BV785 fluorochrome respectively, according to the manufacturer's protocol (Microscale Protein Labeling Kit, Thermo Fisher Scientific) prior to use in flow cytometry. Ig heavy and light chain sequences were synthesized and cloned into IgG1 heavy and kappa backbone expression vectors (GenScript). Antibodies were expressed recombinantly by transient transfection of Ig expression plasmids in Expi293 cells with ExpiFectamine (Thermo Fisher Scientific). Supernatant was harvested 5 days post transfection and antibody was purified by using Protein A Sepharose (Thermo Fisher Scientific). Antibodies 1G01[15], 1G05[40], 2E01[40], FNI9[17], DA03E17[16], and VRC01[37] were produced and purified similarly and used as controls in this study.

### Cell lines

MDCK-SIAT1-PB1 cells[27] used for virus growth inhibition assay, IRINA assay, synergistic assay and virus growth were maintained in DMEM supplemented with 10% FBS, geneticin (1 mg ml$^{-1}$) and puromycin (0.25 µg ml$^{-1}$) at 37 °C with 5% $CO_2$ as described previously[27]. Expi293 cells (Thermo Fisher Scientific, Catalog A14527) were cultured in Expi293 Expression Medium (Life Technologies) agitating at 120 rpm with 8% $CO_2$ at 37 °C.

### Influenza viruses

Rescue and propagation of replication-restricted reporter (R3) and rewired replication-restricted reporter (R4) influenza viruses used in this study was described previously[27]. Briefly, viruses (i.e., H1N1 A/New York/146/2000, H1N1 A/California/7/2009 and H1N1 A/Michigan/45/2015, H2N2 A/Singapore/1/1957, H5N1 A/Vietnam/1203/2004 (clade 1), H5N1 A/Chile/25945/2023 (clade 2.3.4.4b), H5N1 A/Texas/37/2024 (clade 2.3.4.4b), H5N1 A/Victoria/149/2024 (clade 2.3.2.1a), H5N1 A/Cambodia/NPH230032/2023 (clade 2.3.2.1e), and B/Colorado/06/2017) used in this study are R3 or R4 viruses[27] in which PB1 segment was

modified to encode a fluorescent reporter. B/Malaysia/2504/2017 used for the ELLA assay was authentic virus same as the challenge virus.

## Recombinant protein expression and purification

Recombinant NA proteins were expressed by transient transfection of expression vectors in Expi293 cells using ExpiFectamine 293 Transfection Kit (ThermoFisher Scientific). Supernatants were harvested 5 days post-transfection and centrifuged at $3700 \times g$ to remove cell debris. Clarified and filtered supernatants were incubated for 2 h at room temperature with $Ni^{2+}$ Sepharose High-Performane histidine-tagged protein purification resin (Cytiva). Bound protein was eluted using 50 mM Tris pH 8.0, 0.5 M NaCl, 300 mM imidazole. After immobilized metal affinity chromatography by Ni-Sepharose, proteins were further purified by size exclusion chromatography into phosphate-buffer saline (PBS) using a Superdex 200 Increase 10/300 column (Cytiva)[18]. No additional $Ca^{2+}$ was added during protein expression and/or purification.

## Enzyme-linked immunosorbent assay (ELISA)

ELISA was performed as described previously[42]. Briefly, MaxiSorp™ ELISA plates (Nunc) were coated with 2 μg ml[-1] of recombinant NA protein, incubated overnight at 4 °C and washed the next day. MAbs were diluted to 0.1 mg ml[-1] and serial diluted. HRP-conjugated secondary antibody (anti-human IgG, Southern Biotech,Cat# 2040-05) was used and plates were developed with KPL TMB substrate. Absorbance was measured at 450 nm.

## Enzyme-linked lectin assay (ELLA) and NA inhibition assay

Ninety-six-well high-binding ELISA plates were coated with 100 μl of fetuin at 25 μg ml[-1] in a coating buffer (KPL coating solution; SeraCare), sealed and stored at 4 °C overnight or until use. Plates were then washed three times with PBS-T (PBS + 0.05% Tween-20). A two-fold serial dilution of R3 influenza virus was prepared in a separate 96-well plate with a starting dilution of 1/10 in sample diluent (1% BSA in PBS-T). Fifty μl of each virus dilution was added to the washed plate, sealed and incubated overnight at 37 °C. The plates were washed six times with PBS-T and peanut agglutinin (PNA) conjugated HRP (Sigma) at a concentration of 1 μg ml[-1] was added to each plate, incubated for 2 h at room temperature and washed. The plates were developed with SigmaFast OPD in the dark at room temperature and the reaction was stopped after 10 min with 1 N sulfuric acid. Finally, plates were read at a wavelength of 490 nm with a microtiter plate reader. The data were analyzed with Excel (Microsoft) and Prism v10 (GraphPad), and the inhibitory concentration ($IC_{50}$) was calculated.

To perform the NA inhibition assay, the plates were coated and blocked as described above. In a separate plate mAbs were diluted to a starting concentration of 100 μg ml[-1], serially two-fold diluted in sample diluent. Fifty μl of diluted virus was added to each well into the fetuin-coated plate. The same volume of diluted mAb was then added to the plate. Plates were covered with a plate sealer, gently mixed by tapping sides of plates and incubated overnight at 37 °C. After incubation plates were further treated as described above. The data were analyzed and the $IC_{50}$ was calculated as above. A full (100%) and 0% inhibition values were determined from wells with virus without mAb and wells without virus, respectively.

## Influenza replication inhibition neuraminidase-based assay (IRINA)

IRINA assay was performed as described elsewhere[42]. Briefly, viruses were diluted in Opti-MEM (Gibco) and added to Opti-MEM into a 96-well plate and incubated for 1 h at 37 °C. MDCK-SIAT1-PB1 cells were added at a concentration of $1 \times 10^6$ cells ml[-1] to each well containing the diluted virus. Virus-cell mixture was added to each of 384 wells in quadruplicate and incubated for 20–24 h at 37 °C with 5% $CO_2$. Next

day, mAbs were diluted in the assay buffer (NA-Fluor Influenza Neuraminidase Assay Kit; Thermo Fisher Scientific) at a starting concentration of 50 μg ml[-1]. Culture supernatant was removed from the 384-well plate and replaced with diluted mAbs. The plates were incubated for 1 h at 37 °C with 5% $CO_2$, and assay substrate (NA-Fluor Influenza Neuraminidase Assay Kit; Thermo Fisher Scientific) was added to each well and incubated for another 1 h at 37 °C with 5% $CO_2$ and reaction was stopped by adding stop solution (NA-Fluor Influenza Neuraminidase Assay Kit; Thermo Fisher Scientific). Plates were read using an excitation wavelength range of 350 nm to 365 nm and an emission wavelength range of 440 nm to 460 nm. Control wells without the virus infection were used for background subtraction.

## Virus inhibition assay

In a 96-well plate, the mAbs were diluted to a starting concentration of 50 μg ml[-1], serially three-fold diluted in Opti-Mem (Gibco). The viruses were diluted in Opti-MEM supplemented with TPCK-treated trypsin at a concentration of 2 μg ml[-1]. Equal volume (45 μl) of mAb and titrated virus were incubated for 1 hr at 37 °C. MCDK-SIAT1-PB1 cells[27] were trypsinized, washed once with PBS and resuspended in Opti-MEM. MDCK-SIAT-PB1 cells (30,000 cells) were added to each well containing mAb and virus. Each mixture was added in quadruplicate to a 384-well plate and incubated for 4–5 h. After incubation, supernatant was aspirated and wells were replenished with 50 μl of diluted mAb in Opti-MEM with TPCK-treated trypsin at a final concentration of 1 μg ml[-1]. Plates were incubated for an additional 28–48 h and fluorescent objects were counted with Celigo plate reader (Revity) manufactured with customized red filter (EX 540/80 nm, DIC 593 nm, and EM 593/LP nm) using Target1 protocol. Results were analyzed and visualized in GraphPad Prism. Wells with zanamivir or wells without TPCK-treated trypsin were used as controls.

## Biolayer interferometry BLI

BLI experiments were performed by using the Octet HTX instrument (Sartorius). HIS1K biosensors (Sartorius) were hydrated in PBS prior to use. Recombinant NA proteins were immobilized on hydrated HIS1K biosensors through their hexahistidine tag at their N-termini. After brief (60 s) equilibration in an assay buffer (PBS containing 1% BSA) the biosensors were dipped into NCS.1.x mAbs for 600 s to measure association. Biosensors were then dipped in the assay buffer to allow NCS.1.x mAbs to dissociate from NA for 600 s. All assay steps were performed at 30 °C with agitation set at 1000 rpm. Data analysis was done with the Octet analysis software (version 12).

## CryoEM sample preparation

For both Influenza NA samples, prior to freezing, 5 μM of NA of A/California/04/2009 (CA09, H1N1) or A/shorebird/Delaware Bay/309/2016 (DB16, H10N5) was allowed to co-incubate at 4 °C for 30 min (NCS.1.1/CA09 sNAp) or 45 min (NCS.1/DB16) with 15 μM (i.e., a 3-fold molar excess relative to each NA monomer) of fab in 150 mM NaCl, 25 mM Tris (pH 7.5) buffer. To prepare cryoEM sample grids for the bound protein-protein complex, 2 μl of calculated 0.25 mg ml[-1] NA was applied to glow-discharged Quantifoil R 2/2 300 mesh copper grids overlaid with an additional thin layer of carbon. Vitrification was performed on a Mark IV Vitrobot at 22 °C at 100% humidity, with a wait time of 5 s, a blot time of 6 s, and a blot force of 1 before being immediately plunged frozen into liquid ethane. The sample grids were clipped following standard protocols before being loaded into a ThermoFisher Glacios 200 kV transmission electron microscope for imaging.

## CryoEM data collection

NCS.1.1/CA09 sNAp and NCS.1/DB16 NA data were collected automatically using either Leginon[52,53] or SerialEM[54] and used to control a ThermoFisher Glacios 200 kV TEM equipped with a standalone K2

Summit direct electron detector and operating in counting mode. Random defocus ranges spanned between −0.8 and −2.0 μm using stage move, with either one-shot per hole and a single hole per stage move, or 9 holes per stage move with one shot per hole. Altogether, 2212 and 1623 movies were recorded with a pixel size of 0.84 Å and 1.16 Å with a total dose of 50.0 and 59.7 e⁻/Å2 for the NCS.1.1/CA09 sNAp and NCS.1/DB16 NA complexes, respectively.

## CryoEM data processing

All data processing was carried out in cryoSPARC[55,56]. The video frames were aligned using Patch Motion with an estimated B factor of 500 Å2. The maximum alignment resolution was set to 5. Defocus and astigmatism values were estimated using the patch CTF estimation with the amplitude contrast set to 0.07.

For the CA09 sNAp in complex with NCS.1.1, data was initially processed in cryoSPARC Live[57] with blob picking parameters set to a particle size range of 150–200 Å and an extraction box size of 420 pixels. This yielded 623,928 particles, which were subjected to 2D classification. The most well-defined 206,461 particles were then used as templates for a round of template picking. Re-extraction was done with a 460-pixel box size, followed by another round of 2D classification into 100 classes, with three iterations, 50 online-EM iterations, and a batch size of 200 particles per class. Per-particle scale minimization was enabled. The best 419,306 particles from this round were exported to cryoSPARC for further processing. 3D ab initio reconstruction was performed with three classes, using 250,000 particles and C4 symmetry. The best resulting map was used as an input model for 3D heterogeneous refinement, split into four classes with C1 symmetry and hard classification. All four resulting maps demonstrated NCS.1.1 bound to all four protomers of the sNAp of CA09 N1. A total of 370,923 particles from classes with resolutions better than 6 Å were refined using non-uniform refinement with C4 symmetry and per-particle defocus optimization, yielding a map with an estimated global resolution of 2.73 Å. Next, particles were split by exposure group and subjected to global CTF refinement with two iterations, correcting for spherical aberration, beam tetrafoil, and anisotropic magnification. Another round of non-uniform refinement with C4 symmetry and defocus optimization improved the map resolution to 2.50 Å. These particles were then processed using reference-based motion correction, followed by non-uniform refinement with C4 symmetry, yielding an improved resolution of 2.36 Å.Subsequent non-uniform refinement, including per-particle scale minimization, further improved the resolution to 2.35 Å. Incorporating Ewald sphere correction and per-particle scale minimization in another round of refinement brought the map resolution to 2.34 Å. A final round of global CTF refinement, along with non-uniform refinement with C4 symmetry and minimization over per-particle scale, corrected for tetrafoil and anisotropic magnification but not spherical aberration. Ewald sphere correction in the positive direction resulted in a final map resolution of 2.29 Å.

For DB16 NA in complex with NCS.1, a total of 255,414 particles were initially picked using Blob Picker in a reference-free manner and extracted with a 320 Å box size. A first round of reference-free two-dimensional (2D) classification was performed in cryoSPARC using 100 classes, with a maximum alignment resolution of 6 Å. The best classes, containing 98,470 particles, were used for 3D ab initio reconstruction with both C1 and C3 symmetry operators. These same 2D classes were low-pass filtered to 20 Å and used as templates for a second round of particle picking via Template Picker, yielding 286,775 particles, which were again extracted with a 320-pixel box size. A second round of 2D classification was performed, and the best 136,494 particles were submitted for 3D heterogeneous refinement into three classes using C1 symmetry and hard classification. Particles corresponding to maps demonstrating NCS.1 binding to all four NA binding sites were refined using 3D non-uniform refinement, employing the initial model from the previous C4 3D ab initio job, resulting in an initial map resolution of

3.58 Å. Particles were re-centered and re-extracted before running local motion correction with a maximum alignment resolution of 3 Å and a 320-pixel box size. A subsequent homo-refinement step yielded a map at 3.60 Å, which was then used as the initial model for downstream non-uniform refinement. This stage used inner and outer window radii of 0.85 and 0.99, with C4 symmetry applied. Low-pass filtering was set to 20, GSFSC split resolution to 15, and per-particle scale minimization was enabled. Per-particle defocus parameters were optimized with a reduced defocus search range of 1,000 Å, improving the global resolution to 3.38 Å. Reference-based motion correction was then performed, followed by another round of 3D non-uniform refinement, resulting in a map with an estimated global resolution of 3.37 Å and improved density features. Global CTF correction was applied to particles from this map, followed by another non-uniform refinement with C4 symmetry and further per-particle scale and defocus optimization (defocus search range reduced to 500 Å), yielding a final global resolution of 3.35 Å.

For the 3 Fab-bound map, a similar processing pipeline was used following the 3D heterogeneous refinement step, utilizing 75,643 particles that were only bound to three of the four catalytic NA pockets. This resulted in a final map with an estimated global resolution of 4.25 Å. DeepEMhancer[58], with the HighRes learning model, was used to locally sharpen regions of the final maps.

Local resolution estimates for all three cryoEM structures were calculated in cryoSPARC using an FSC value of 0.143. 3D maps for the two half-maps, the final unsharpened map and the final sharpened map were deposited in the Electron Microscopy Data Bank under accession number EMD-48102, EMD-70264, EMD-48093, and EMD-48101 for NCS.1.1/CA09 sNAp [C4], NCS.1.1/CA09 sNAp [C1], NCS.1/DB16 NA (4 fabs), and NCS.1/DB16 NA (3 fabs), respectively.

## CryoEM model building and validation

For the structure of NCS.1.1/CA09 sNAp, the model was initially built automatically using ModelAngelo upon providing sequence information for both the sNAp and the NCS.1.1 fab. In contrast, the structure of 1G01 Fab (PBD: 6Q23) was used as an initial reference for building the final cryoEM structure of NCS.1/DB16 NA. All models were manually edited and trimmed using Coot[59–61]. We then further refined each structure using a combination of Rosetta using density-guided protocols and ISOLDE[62]. This process was repeated iteratively until convergence and high agreement with the map was achieved. Multiple rounds of relaxation and minimization were performed on each structure, which was manually inspected for errors each time. Residues with poor density were truncated to their corresponding Cβ carbons. Privateer[63] was used to verify correct glycan geometry. Phenix real-space refinement was subsequently performed as a final step before the final model quality was analyzed using MolProbity[64]. Figures were generated using either UCSF Chimera[65] or UCSF ChimeraX[66]. The final structures was deposited in the PDB under accession numbers 9EJF, 9O9V, 9EIT, and 9EJE for NCS.1.1/CA09 sNAp [C4], NCS.1.1/CA09 sNAp [C1], NCS.1/DB16 NA (4 fabs), and NCS.1/DB16 NA (3 fabs), respectively.

## CryoEM water placement and validation

Initial water molecules were placed into cryoEM density using Phenix.douse on the C4 symmetry–refined map of the NCS.1.1/N1-sNAp complex, followed by manual validation and adjustment in Coot. Additional water molecules were manually added in regions where unmodeled density could be reasonably attributed to water. These placements were further refined through molecular dynamics with flexible fitting in ISOLDE, allowing the water molecules to relax into energetically favorable positions that better matched the local density.

To validate these placements, the structure from the C4 map was docked into a C1 symmetry–refined map of the same complex, and water positions were reassessed using CheckWaters in ChimeraX. A density threshold of 0.2 was applied in ChimeraX, meaning

that waters were only retained if they were supported by map values at or above this contour level. Waters below this threshold were removed (165 in total), as they were likely attributable to noise or weak occupancy averaged out in the C4 map. For the C4 map, waters that were retained following these steps were kept for the C4-refined PDB structure.

For the C1 map, Phenix.douse was rerun with a deliberately permissive map_threshold_scale of 0.7 and sphericity_filter set to false to ensure comprehensive sampling of potential water sites. The resolution parameter was set to 2.5 Å, consistent with the global resolution of the map (with local resolution at the NCS.1.1–N1 interface being higher). It is important to note that the Phenix.douse threshold governs initial water placement, while the ChimeraX threshold is used post hoc to further assess whether placed waters are supported by density.

Following automated placement in the C1 map, manual inspection was conducted for all water molecules, with particular focus on those at the NCS.1.1–N1 interface. Water placements were also compared across all protomers in the C1 asymmetric unit, revealing that many waters were consistently found in the same positions across symmetry-independent copies. This internal consistency, along with strong map support and high resolution, provides high confidence that the retained waters represent genuine structural features rather than noise. Although the C1 map has a slightly lower global resolution than the C4 map, the authors recommend using the C1 map for future analyses of water molecule contributions along the interface between NCS.1.1 and the N1-sNAp, as it was subjected to a more detailed analysis of water placement and variance across individual protomers.

### Statistical analysis and experimental reproducibility

All in vitro experimental data shown except for B cell sorting and cryoEM structure determination were representative of at least two independent experiments. Conclusions from the repeated experiments that are not shown in the manuscript are similar to those from the shown experiments. The sample size for in vivo experiments was determined based on the expected heterogeneity of the samples, the significance threshold (chosen at 0.05), the expected or observed difference, as well as previous publications and our pilot studies. The chosen sample size in each experiment is sufficient to generate statistically significant results. In all experiments, unless otherwise indicated, data are shown as mean with all data points or group mean $\pm$ SD, and statistical analyses were performed using GraphPad Prism (v10). Mantel-Cox test was used to compare Kaplan–Meier curves with Bonferroni correction applied. $P$-values less than or equal to 0.05 (0.0033 after Bonferroni correction) were considered significant.

### Reporting summary

Further information on research design is available in the Nature Portfolio Reporting Summary linked to this article.

## Data availability

All materials generated in this study will be made available with a material transfer agreement by the corresponding authors (N.P.K., and M.K.) upon request. Structural models and density maps reported in this study are deposited in the Protein Data Bank and Electron Microscopy Data Bank under accession numbers 9EJF (NCS.1.1/CA09 sNAp [C4]), 9O9V (NCS.1.1/CA09 sNAp [C1]), 9EIT (NCS.1/DB16 NA [4 fabs]), and 9EJE (NCS.1/DB16 NA [3 fabs]); and EMD-48102 (NCS.1.1/CA09 sNAp [C4]), EMD-70264 (NCS.1.1/CA09 sNAp [C1]), EMD-48093 (NCS.1/DB16 NA [4 fabs]), and EMD-48101 (NCS.1/DB16 NA [3 fabs]), respectively. PDB entries 8G3P and 6Q23 are used for structural comparisons. Source data are provided as a Source Data file. Source data are provided with this paper.

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

## Acknowledgements

The authors thank A. Widge, M. Crank, J. Ledgerwood, L. Dropulic at the VRC, and the VRC 200 study participant and study team for human samples; D. Ambrozak and flow cytometry core of the VRC for help with cell sorting; S. Lakdawala (Emory University) for providing sequence of the H5N1 2.3.4.4b A/dairy cow/24008749001/2024 virus; W. Paul Duprex (University of Pittsburgh Center for Vaccine Research) for paying for the plasmids used for rescuing the H5N1 2.3.4.4b A/dairy cow/24008749001/2024 virus; and members of the VRC Influenza Program for helpful discussion. Additionally, we thank K. VanWormer, H. Nunez-Ortega, R. Ticzon, A. Dubief, and G. Ruth for providing laboratory support at the Institute for Protein Design, as well as L. Goldschmidt, P. Vecchiato, and B. Faezov for technical support. Molecular graphics and analyses performed with UCSF ChimeraX, developed by the Resource for Biocomputing, Visualization, and Informatics at the University of California, San Francisco, with support from National Institutes of Health R01-GM129325 and the Office of Cyber Infrastructure and Computational Biology, National Institute of Allergy and Infectious Diseases (NIAID). This work was supported in part by the Intramural Research Program of the National Institutes of Health (NIH) (M.K.); a generous gift from Open Philanthropy (N.P.K., A.J.B.); and the Audacious Project at the Institute for Protein Design (N.P.K., A.J.B.). An NIH award (UC7AI180311) from NIAID supported the operations of The University of Pittsburgh Regional Biocontainment Laboratory within the Center for Vaccine Research. NHMRC grants GNT1173433 and GNT2009711 (A.K.W., H.-X.T.) supported the study conducted in the University of Melbourne. The contributions of the NIH authors were made as part of their official duties as NIH federal employees, are in compliance with agency policy requirements, and are considered Works of the United States Government. However, the findings and conclusions presented in this paper are those of the authors and do not necessarily reflect the views of the NIH or the U.S. Department of Health and Human Services.

## Author contributions

Conceptualization—J.L., A.J.B., D.E., M.K; Methodology—J.L., A.J.B., L.N., R.A.G.; Investigation—J.L., A.J.B., L.N., C.J.W., E.L.W., J.R., C.Y., A.C., H-X.T., T.H.T.D., M.R., V.L.S.; Formal Analysis—J.L., A.J.B L.N., R.A.G.; Resources—V.L.S., D.S.R.; Data Curation—A.J.B.; Writing—Original Draft —J.L., A.J.B.; Writing—Review and Editing—J.L., A.J.B., L.N., R.A.G., N.P.K., M.K.; Supervisions—A.B.M., S.F.A., A.K.W., D.S.R., N.P.K., M.K.; Funding Acquisitions—B.S.G., N.P.K., M.K.

## Funding

## Competing interests

J.L., D.E., B.S.G., N.P.K., and M.K. are named inventors of a patent application describing engineered influenza neuraminidase antigens under publication number WO/2021/178621 filed by the University of Washington and the Department of Health and Human Services, USA. The remaining authors declare no competing interests.

## Additional information

[1]Vaccine Research Center, National Institute of Allergy and Infectious Diseases, National Institutes of Health, Bethesda, MD 20892, USA. [2]Institute for Protein Design, University of Washington, Seattle, WA 98195, USA. [3]Department of Biochemistry, University of Washington, Seattle, WA 98195, USA. [4]Department of Immunology, Center for Vaccine Research, University of Pittsburgh, Pittsburgh, PA, USA. [5]Graduate Program in Molecular and Cellular Biology, University of Washington, Seattle, WA 98195, USA. [6]Department of Microbiology and Immunology, Peter Doherty Institute for Infection and Immunity, University of Melbourne, Melbourne VIC 3000, Australia. [7]These authors contributed equally: Julia Lederhofer, Andrew J. Borst. ✉e-mail: neilking@uw.edu; kanekiyom@nih.gov

