## [Transparent Peer Review file · Nature Communications]

Structural Convergence and Water-Mediated Substrate Mimicry Enable Broad Neuraminidase Inhibition by Human Antibodies

Corresponding Author: Dr Masaru Kanekiyo

Version 0:

Reviewer comments:

Reviewer #1

(Remarks to the Author)

This manuscript describes an approach to isolate broadly reactive antibodies to subtypes of the influenza neuraminidase protein. The authors isolated B cells from PBMCs of an individual infected with H3N2 some years previously. The B cells were probed with fluorochrome labeled stabilized N4 and N5 NAs, and single memory cells bound to the probes were isolated by flow cytometry. The antigen binding regions of the antibodies encoded by these B cells were cloned and inserted into a backbone antibody expression vector. Six antibodies were prepared and the antibodies characterized by binding to stabilized NA from different subtypes. The antibodies demonstrated a rather broad binding pattern, binding to N4, N5, N1, and even an NA from an influenza B virus. The authors also determined the protection afforded by antibodies to infections with different subtypes of influenza, including H1N1, H5N1, and an influenza B virus as well as neuraminidase inhibitory activity of the antibodies using two different substrates.

Using cryo-EM the authors determined the structure of the antibodies complexed with different NAs. Four antibodies bound to the conserved neuraminidase active site in different stabilized NA. Their structures indicated that the antibodies all bound to NA active site through sialic acid mimicry. They also compared binding of their new Ab with those of previously isolated cross reacting antibodies. They conclude that in all the broadly reacting antibodies, conserved motifs contributed to the breadth of the binding.

The authors also conclude that not only sialic acid mimicry contributed to the breadth of activity but also water assisted substrate mimicking aids in the breadth of the binding.

The protocol used to generate the broadly reactive antibodies is quite powerful and likely could be used for antigenically different variants of other viruses. The analyses of the antibodies generated here are also quite comprehensive.

Issues to be considered:

Analyses of the effects of water molecules in the conformation of an active site should be explained in more depth. For example, line 21—how was modeling of the network of water molecules accomplished with cryo-EM.

Lines 36-41—precisely how was it concluded that “breadth of the mAb was due partly to water-mediated interactions that adapt to subtype specific residue variations in the catalytic pocket”. Other mechanisms seem possible. Is it possible to have a negative control for water mediated interactions? Is it possible to remove the water in molecular dynamic simulations to determine the effects on the binding of antibodies to the active sites of different NAs?

The statistical significance of differences in groups shown in Figures 5 and 6 in main body of text and in extended Figures 1 and 5 are not indicated. The Materials and Methods describe such analyses, but p values are not in the figures nor the figure legends.

Reviewer #2

(Remarks to the Author)

Lederhofer et al focused on the characterization of six human monoclonal antibodies that target the conserved catalytic site of influenza neuraminidase, named NCS.1.x mAbs. The NCS.1.x mAbs were derived from memory B cells that were isolated using unrelated NAs as the bait from a donor who had the H3N2 infection at 2015. All NCS.1.x mAbs broadly bound to group 1 N1s and N5 and some of them bound to Victoria lineage flu B NA. The study of NCS.1.1 in complex with N1-CA09-

sNAP and NCS.1 in complex with N5-DB16 using cryoEM revealed that antibodies recognize the catalytic pocket of NA via substrate mimicry and antibodies engage the NA via a distinctive binding pattern, comparing to another receptor mimicry antibody FNI9. The structure analysis also showed that the water molecules facilitate the interactions between the heavy chain CDR3 region of antibody NCS.1 & NCS.1.1 and the NA. The NSC .1.x mAbs had variable activities in inhibiting the NA of N1s, N2 and Victoria lineage flu B tested. The NSC .1.x mAbs were also tested for prophylaxis against the lethal challenge of H1N1 CA09, Victoria B MA04, H5N1 VN04 and H5N1 TX24 in a murine model.

Overall, the work is interesting and provides certain information about anti-influenza NA antibodies. The NCS.1.x antibodies recognize conserved catalytic site of influenza NA via receptor mimicry, which in part explains their binding breadth with N1, N5 and flu B tested in the study. However, limited NA inhibition data and lack of detailed epitope and sequence alignment of critical residues restrict the relevance of the work and raise the concern about the broad NA inhibition by these antibodies. The modest efficacy against H5N1 jeopardizes the application of these antibodies and contradicts the claim of prophylactic option for pandemic preparedness.

Though NCS.1.x mAbs more likely react with group 1 NAs, only seasonal N1s of CA09 and MI15 and avian N1s of three H5N1s were included in the functional assay. The authors may consider to include earlier and latest seasonal N1s (G248K/R/E in earlier or recent N1s, K432E in recent N1s can affect catalytic site-targeting anti-N1 antibodies). The functional data against N1 of recently emerging 2.3.2.1c H5N1 is not seen in the study either. N5 was used to isolate the B cells but only binding data with N5-DB16 was provided.

While the inhibitory activities against N1s tested in the study were comparable, detailed epitope analysis and contact residues alignment may help to explain substantial variations of activities against N1s, N2 and Vic flu B. The above concern is also raised when the antibody fails to maintain activities across a wide range of N1s.

Some other comments are as follows,

1. Please provide the antibody numbering scheme.
2. It is a bit weird to isolate B cells against group 1 NAs from the H3N2-exposed donor. I was wondering if mixed infection or recent infection with H1N1 is excluded for donor A. Time point 1 sample was collected two to six weeks post-confirmed influenza positive PCR test for donor A, recent H1N1 infection between two time points may occur in the flu season.
3. Is Fig.1a better to be referred in line 19 rather than line 17 at page 3?
4. Line 36 at page 3, N5 DE16. Is this a typo?
5. In Fig.1c, the NCS.1.2 rather than NCS.1.1 bind to N2 WI05. Line 29 at page 3, is this a type?
6. Lines 8-10 at page 14, is the figure legend of Fig.1d and Fig.1e inversely indicated?
7. Is there high-resolution data for NCS.1.1-, NCS.1.2-, and NCS.1.3-bound NA of B/Vic CO17 (Fig.4g)?

Reviewer #3

(Remarks to the Author)

Lederhofer, Borst et al., report here on monoclonal antibodies (mAbs) that target the catalytic site (NCS) of the influenza virus receptor-destroying enzyme, neuraminidase.

The authors isolated, and comprehensively characterize these mAbs and showed that they were broadly against several NAs, including N1, N5, and B NA. Mostly notably, the cryo-EM structures of NCS mAbs in complex with NAs revealed that the amino acid residues at the tips of the mAbs' CDR3 loops mimic the enzyme's substrate, sialic acid. Given the relatively high resolution of their cryo-Em reconstructions, the authors were able to identify a network of water molecules involved in the solvation of the antibody epitope and arranged in a similar fashion to that of SIA-bound NA structure.

This is a high-quality manuscript and my suggestions on improvement are minor:

- 1 - It appears that many, if not all, glycans have issues with stereochemistry. I suggest using Privateer to validate and correct the conformers.
- 2 - Why are hydrogens included in the final atomic coordinates? The resolution does not allow to visualize them, so they should only be used in the refinement steps to minimize clashes, but removed from the final model.
- 3 - Could you elaborate on the cryo-EM data processing? It looks like the water molecules were overzealously placed, i.e., many are not visible at medium and higher map contours. Did you use phenix.douse, for example, to check? While on that point, I wonder how many waters get averaged out simply by enforcing the C4 symmetry? Are they all also present in the C1 refined structure?
- 4 - The methods section could be more detailed, especially with NA purification etc. For example, Ca²⁺ is present in the structure but there is no mention of calcium in the purification or grid preparation protocols.
- 5 - You could consider IMGT numbering for the mAbs, instead of Kabat.
- 6 - (d) and (e) are inverted in Figure 1 legend

P.S. Thank you for sending the primary cryo-EM maps - that made my job much easier!

Version 1:

Reviewer comments:

Reviewer #2

(Remarks to the Author)

The authors have appropriately responded to and addressed points raised in my and reviewer #1's original reviews.

A minor issue:

Page 25, line 24: DE16, a typo?

Reviewer #3

(Remarks to the Author)

All my initial concerns have been addressed in the revision.

Point-by-point response to reviewers' comments:

Summary of the findings

Reviewer #1 (Remarks to the Author):

This manuscript describes an approach to isolate broadly reactive antibodies to subtypes of the influenza neuraminidase protein. The authors isolated B cells from PBMCs of an individual infected with H3N2 some years previously. The B cells were probed with fluorochrome labeled stabilized N4 and N5 NAs, and single memory cells bound to the probes were isolated by flow cytometry. The antigen binding regions of the antibodies encoded by these B cells were cloned and inserted into a backbone antibody expression vector. Six antibodies were prepared and the antibodies characterized by binding to stabilized NA from different subtypes. The antibodies demonstrated a rather broad binding pattern, binding to N4, N5, N1, and even an NA from an influenza B virus. The authors also determined the protection afforded by antibodies to infections with different subtypes of influenza, including H1N1, H5N1, and an influenza B virus as well as neuraminidase inhibitory activity of the antibodies using two different substrates.

Using cryo-EM the authors determined the structure of the antibodies complexed with different NAs. Four antibodies bound to the conserved neuraminidase active site in different stabilized NA. Their structures indicated that the antibodies all bound to NA active site through sialic acid mimicry. They also compared binding of their new Ab with those of previously isolated cross reacting antibodies. They conclude that in all the broadly reacting antibodies, conserved motifs contributed to the breadth of the binding.

The authors also conclude that not only sialic acid mimicry contributed to the breadth of activity but also water assisted substrate mimicking aids in the breadth of the binding.

The protocol used to generate the broadly reactive antibodies is quite powerful and likely could be used for antigenically different variants of other viruses. The analyses of the antibodies generated here are also quite comprehensive.

Response:

We thank the reviewer for their positive assessment of our manuscript. Our response to specific comments follow.

Issues to be considered:

Analyses of the effects of water molecules in the conformation of an active site should be explained in more depth. For example, line 21—how was modeling of the network of water molecules accomplished with cryo-EM.

Response:

We appreciate the reviewer's request for further clarification on how water-mediated interactions contribute to the breadth of NCS.1.x mAbs. We have expanded on our discussion of this point in the revised manuscript, explaining how our structural analysis demonstrates that water molecules stabilize the CDRH3 loop and bridge gaps along the binding interface, allowing for effective engagement with the NA catalytic pocket despite sequence variations. Additional

details for how we modeled of the water network are now described in depth in a new Methods section of our revised manuscript.

Page 4, lines 25–28:

“To ensure accurate water placement, we carried out a systematic validation pipeline (see Methods), confirming that all modeled waters were supported by clear density in both C4 and C1 reconstructions of the complex (Extended Data Fig. 3).”

Pages 27-28, lines 34–19:

“CryoEM water placement and validation

Initial water molecules were placed into cryoEM density using Phenix.douse on the C4 symmetry–refined map of the NCS.1.1/N1-sNAp complex, followed by manual validation and adjustment in Coot. Additional water molecules were manually added in regions where unmodeled density could be reasonably attributed to water. These placements were further refined through molecular dynamics with flexible fitting in ISOLDE, allowing the water molecules to relax into energetically favorable positions that better matched the local density.

To validate these placements, the structure from the C4 map was docked into a C1 symmetry–refined map of the same complex, and water positions were reassessed using CheckWaters in ChimeraX. A density threshold of 0.2 was applied in ChimeraX, meaning that waters were only retained if they were supported by map values at or above this contour level. Waters below this threshold were removed (165 in total), as they were likely attributable to noise or weak occupancy averaged out in the C4 map. For the C4 map, waters that were retained following these steps were kept for the C4-refined PDB structure.

For the C1 map, Phenix.douse was rerun with a deliberately permissive map_threshold_scale of 0.7 and sphericity_filter set to false to ensure comprehensive sampling of potential water sites. The resolution parameter was set to 2.5 Å, consistent with the global resolution of the map (with local resolution at the NCS.1.1–N1 interface being higher). It is important to note that the Phenix.douse threshold governs initial water placement, while the ChimeraX threshold is used post hoc to further assess whether placed waters are supported by density.

Following automated placement in the C1 map, manual inspection was conducted for all water molecules, with particular focus on those at the NCS.1.1–N1 interface. Water placements were also compared across all protomers in the C1 asymmetric unit, revealing that many waters were consistently found in the same positions across symmetry-independent copies. This internal consistency, along with strong map support and high resolution, provides high confidence that the retained waters represent genuine structural features rather than noise.”

Lines 36-41—precisely how was it concluded that “breadth of the mAb was due partly to water-mediated interactions that adapt to subtype specific residue variations in the catalytic pocket”. Other mechanisms seem possible. Is it possible to have a negative control for water mediated interactions? Is it possible to remove the water in molecular dynamic simulations to determine the effects on the binding of antibodies to the active sites of different NAs?

Response:

We thank the reviewer for this insightful comment and agree that water molecules are not the sole contributors to antibody breadth. Other factors—such as CDR flexibility, CDRH3 length, and contacts both within and outside the CDRH3, including the conserved RD/DR motif described in this and the accompanying manuscript—likely play important roles as well. In the present study, we emphasize water-mediated interactions because they are clearly resolved in our high-resolution cryoEM structures and because other features contributing to breadth in NA catalytic site-targeting antibodies have been well documented by others as well as the accompanying manuscript by Jo et al. (ref. 41) and a recent publication by Madsen et al. (ref. 40). These water-mediated interactions align with established principles of protein-protein recognition, in which water often bridges contacts in regions that are sterically or chemically constrained (reviewed by Janin, *Structure*, 1999), and have been associated with breadth in antibodies targeting HIV and influenza (refs. 36–39).

Although such interactions may have not been described in earlier NA antibody structures due to limited resolution, our data allow us to confidently document them in this context. Therefore, we do not believe molecular dynamics simulations are necessary to further assess their relevance. We now clarify in the revised manuscript that water-mediated contacts likely function in concert with other molecular features, these interactions have been observed in other broadly neutralizing antibodies, and their relative contribution remains an open and important area for future study.

Page 9, lines 6–9:

“Indeed, broadly neutralizing antibodies against HIV-1 (e.g., VRC01-class and PG9) and influenza HA (e.g., C05) utilize water molecules to bridge interactions with structurally diverse antigens, allowing them to accommodate antigenic variation while maintaining high-affinity binding^{36–39}.”

Page 9, lines 12–16:

“While our structural data highlight the role of water molecules in bridging contacts along the binding interface, we recognize that they are likely not the sole determinant of breadth. Other factors, such as CDR loop flexibility, length, and its amino acid composition, and the conserved interactions through the CDRH3 motif with the catalytic site, likely also play major roles^{40–41}.”

The statistical significance of differences in groups shown in Figures 5 and 6 in main body of text and in extended Figures 1 and 5 are not indicated. The Materials and Methods describe such analyses, but p values are not in the figures nor the figure legends.

Response:

We thank the reviewer for noticing the missing statistical significance. We added a table in our supplementary to report statistical significance of the Kaplan-Meier curves, related to main

Figure 6. Data of main Figure 5 and extended Figures 1 and 5 show response curves with mean \pm SD. No statistical analyses were performed to compare curves.

Reviewer #2 (Remarks to the Author):

Lederhofer et al focused on the characterization of six human monoclonal antibodies that target the conserved catalytic site of influenza neuraminidase, named NCS.1.x mAbs. The NCS.1.x mAbs were derived from memory B cells that were isolated using unrelated NAs as the bait from a donor who had the H3N2 infection at 2015. All NCS.1.x mAbs broadly bound to group 1 N1s and N5 and some of them bound to Victoria lineage flu B NA. The study of NCS.1.1 in complex with N1-CA09-sNAp and NCS.1 in complex with N5-DB16 using cryoEM revealed that antibodies recognize the catalytic pocket of NA via substrate mimicry and antibodies engage the NA via a distinctive binding pattern, comparing to another receptor mimicry antibody FNI9. The structure analysis also showed that the water molecules facilitate the interactions between the heavy chain CDR3 region of antibody NCS.1 & NCS.1.1 and the NA. The NSC .1.x mAbs had variable activities in inhibiting the NA of N1s, N2 and Victoria lineage flu B tested. The NSC .1.x mAbs were also tested for prophylaxis against the lethal challenge of H1N1 CA09, Victoria B MA04, H5N1 VN04 and H5N1 TX24 in a murine model.

Overall, the work is interesting and provides certain information about anti-influenza NA antibodies. The NCS.1.x antibodies recognize conserved catalytic site of influenza NA via receptor mimicry, which in part explains their binding breadth with N1, N5 and flu b tested in the study. However, limited NA inhibition data and lack of detailed epitope and sequence alignment of critical residues restrict the relevance of the work and raise the concern about the broad NA inhibition by these antibodies. The modest efficacy against H5N1 jeopardizes the application of these antibodies and contradicts the claim of prophylactic option for pandemic preparedness.

Response:

We thank the reviewer for their kind words and critical review of our manuscript. Our detailed response to the reviewer's comments on NA inhibition, epitope alignment, and activity against H5N1 follow. We also added additional data on H5N1 viruses from clades 2.3.2.1a and 2.3.2.1e (formerly 2.3.2.1c) in the revised Figure 5 in response to the reviewer's request. Regarding the claim about potential pandemic countermeasures, we acknowledge the limitations and agree that our NCS.1.x mAbs are not meant to be the solution for a prophylactic option for future pandemics. We revised our discussion accordingly.

Page 7, lines 14–16:

“Importantly, the ability of these mAbs to effectively suppress the 2.3.4.4b and 2.3.2.1a H5N1 viruses underscores their potential utility in developing effective countermeasures against emerging influenza viruses with pandemic potential.”

Page 8, lines 11–12:

“These mAbs demonstrated robust prophylactic protection in murine models, providing insights that would lead to the development of effective tools for combating influenza infection.”

Though NCS.1.x mAbs more likely react with group 1 NAs, only seasonal N1s of CA09 and MI15 and avian N1s of three H5N1s were included in the functional assay. The authors may consider to include earlier and latest seasonal N1s (G248K/R/E in earlier or recent N1s, K432E in recent N1s can affect catalytic site-targeting anti-N1 antibodies). The functional data against N1 of recently emerging 2.3.2.1c H5N1 is not seen in the study either. N5 was used to isolate the B cells but only binding data with N5-DB16 was provided.

Response:

We thank the reviewer for raising this important question. Regarding the K432E substitution, structural analysis of our NCS.1.x Fab–NA complexes shows that K432 is located distally from the NCS.1.x binding epitope and does not participate in Fab–antigen contacts. This is consistent with our binding and inhibition data, which show no sensitivity to this residue variation (Extended Data Fig. 5).

In contrast, G248 lies within the NCS.1.x binding epitope and is therefore of greater potential concern. While substitutions at this position (e.g., G248K/R/E) would be expected to disrupt binding due to differences in side chain size and charge, our functional data indicate that our antibodies retained activity (Extended Data Fig. 5). Structural comparisons of our two NCS.1.x–NA complexes—one with N1 (G248) and one with N5 (S248)—reveal that the CDRH1 loop undergoes a substantial conformational change to accommodate the G248S substitution in the N5 structure. Notably, the remainder of the CDRH1 epitope is conserved between the two complexes (Extended Data Fig. 5). This rearrangement repositions the loop backbone to maintain epitope engagement and displaces solvent water molecules that mediate binding in the N1-bound complex.

This adaptability underscores the conformational plasticity of the NCS.1.x CDRH1 loop, which contains the glycine- and serine-rich sequence GESISSGGY. This glycine-serine-rich sequence imparts considerable flexibility, allowing the loop to structurally adapt to epitope variation while preserving functional binding. Such plasticity likely accounts for the retained efficacy of NCS.1.x mAbs even in the presence of G248K/R/E substitutions. In addition, we added IRINA data with A/New York/146/2000 H1N1 virus (Figure 5a), which possesses R248. Our NCS.1.x mAbs remain active against this R248-containing virus, further accentuating the tolerability of variations at the position 248 by these mAbs.

Regarding H5N1, we performed additional IRINA and ELLA inhibition assays using H5N1 A/Cambodia/NPH230032/2023 (clade 2.3.2.1e, formerly 2.3.2.1c) and A/Victoria/149/2024 (clade 2.3.2.1a) viruses (Fig. 5e and f). Our NCS.1.x mAbs were highly active against both 2.3.2.1 subclade H5N1 viruses in addition to the 2.3.4.4b H5N1 viruses despite their NAs being substantially different. We also showed NA Inhibition against recombinant N5 DB16 by our NCS.1.x mAbs (Figure 5h).

We added or updated the following text in the revised manuscript to describe these new results to the reader:

Page 6, lines 24–32:

“The G248R mutation of A/New York/146/2000 H1N1 did not impact binding of the NCS.1.x mAbs. This amino acid change is centrally located within the heavy chain footprint of the NCS.1.x mAbs (Extended Data Fig. 5a). Structural analysis reveals that despite its central location, G248 mutations are accommodated by conformational flexibility within the CDRH1 loop of the antibody. The glycine-serine-rich motif of CDRH1 facilitates structural adaptation, allowing the loop to avoid steric clashes introduced by bulkier side chains. In our N5–DB16 complex, which harbors a related G248S mutation, this rearrangement also displaces water molecules from the original binding site while preserving key interactions through alternative hydrogen bonding or van der Waals contacts (Extended Data Fig. 5b,c).”

Page 7, lines 1–16:

“Given the ongoing global panzootic H5N1 outbreak, we next assessed the potential utility of the NCS.1.x mAbs against H5N1 clade 2.3.4.4b, 2.3.2.1a and 2.3.2.1e viruses. Both IRINA and ELLA were performed with A/Chile/25945/2023 (H5N1 2.3.4.4b, CH23), A/Texas/37/2024 (H5N1 2.3.4.4b, TX24), A/Victoria/149/2024 (H5N1 2.3.2.1a, Vic24) and A/Cambodia/NPH230032/2023 (H5N1 2.3.2.1e, Cam23) reporter viruses which were engineered using the HA and NA sequences of isolates from recent human H5N1 cases in Chile²², Texas²³, Victoria²⁴, and Cambodia²⁵, respectively. All NCS.1.x mAbs as well as 1G01 were highly potent against H5N1 viruses, representing three different clades both in IRINA and ELLA (Fig. 5e-f). Other catalytic site-targeting antibodies, including FNI9, 1G05²⁶, 2E01²⁶, and DA03E17¹⁶, were also tested and effectively inhibited NA catalytic activity against various N1 subtypes, including the 2.3.4.4b H5N1 CH23 and TX24 (Extended Data Fig. 6). We then assessed whether NCS.1.x mAbs inhibit in vitro viral propagation of H1N1 CA09 and four H5N1 viruses, including VN04, CH23, TX24, and Vic24. The virus growth inhibition assay utilized reporter influenza viruses and infected cells were monitored over time²⁷. All NCS.1.x mAbs demonstrated inhibitory activity against all tested viruses with similar potency as 1G01 (Fig. 5g). Importantly, the ability of these mAbs to effectively suppress the 2.3.4.4b and 2.3.2.1a H5N1 viruses underscores their potential utility in developing effective countermeasures against emerging influenza viruses with pandemic potential.”

Page 7, lines 16–18:

“We additionally demonstrated that all NCS.1.x mAbs demonstrated ELLA inhibition activity against recombinant N5 DB16 (Fig. 5h), confirming our ELISA binding data (Fig. 1c).”

While the inhibitory activities against N1s tested in the study were comparable, detailed epitope analysis and contact residues alignment may help to explain substantial variations of activities against N1s, N2 and Vic flu B. The above concern is also raised when the antibody fails to maintain activities across a wide range of N1s.

Response:

We agree with the reviewer that while NAI activities against the N1s tested in this study were generally comparable, we observed weaker activity against group 2 and influenza B NAs. As

suggested, we performed an in-depth structural and sequence analysis of the NCS.1.1–N1 interface to define residues involved in subtype specificity. Sequence alignment of the NA contact residues across pre-pandemic seasonal, pandemic, and avian N1s, as well as representative N2 and influenza B strains, revealed that residues S247, N273, and A343 are highly conserved among various N1 NAs but divergent in N2 and influenza B NAs (see below). Structurally, S247 is located near the catalytic pocket and in close proximity to the CDRH3 loop of NCS.1.1. Substitution of S247A as seen in non-N1 NAs may disrupt the local electrostatic milieu or hydrogen bonding interactions critical for Fab engagement. Although N273 is located outside the catalytic site, it is a known N-linked glycosylation site. Loss of this glycan (in N2 or influenza B) could significantly alter the local epitope landscape, potentially impairing antibody recognition. Together, these differences may explain the limited breadth and potency of NCS.1.x mAbs outside of the N1 NAs. Since these observations are speculative and not supported by empirical data, we have decided not to include them in our manuscript.

Some other comments are as follows,

1. Please provide the antibody numbering scheme.

Response:

We used the Kabat numbering scheme for the antibodies. We now include this in Figure 1 caption.

2. It is a bit weird to isolate B cells against group 1 NAs from the H3N2-exposed donor. I was wondering if mixed infection or recent infection with H1N1 is excluded for donor A. Time point 1 sample was collected two to six weeks post-confirmed influenza positive PCR test for donor A, recent H1N1 infection between two time points may occur in the flu season.

Response:

We thank the reviewer for their insightful comment. The patient was neither subtyped nor re-tested for possible reinfection, thus we are not able to exclude the possibility of this donor being co-infected with two subtypes nor reinfected with a second virus between the time points. However, we do see N1-specific B cells at the earlier time point, suggesting the reinfection scenario is unlikely. To note, PBMCs from this donor were also used to isolate B cells and antibodies with N2 NA probes, and the data were published elsewhere (Lederhofer et al., Immunity, 2024). We have added clarifications in the Methods but decided not to discuss highly speculative scenarios.

Page 22, lines 13–14:

“This donor PBMC sample was also used to isolate N2 NA-specific B cells in our previous study⁴².”

3. *Is Fig. 1a better to be referred in line 19 rather than line 17 at page 3?*

Response:

We thank the reviewer for pointing this out. We have added an additional sentence at the beginning of our results section where Fig. 1a is referred to.

Page 3, lines 15–16:

“The functionally and structurally conserved catalytic pocket of NA (Fig. 1a) and its relatively conserved antigenic properties makes it an attractive target for broadly cross-reactive antibodies.”

4. *Line 36 at page 3, N5 DE16. Is this a typo?*

Response:

We thank the reviewer for noticing this error. This is indeed a typo and should have been DB16. We have corrected the mistake.

5. *In Fig. 1c, the NCS.1.2 rather than NCS.1.1 bind to N2 WI05. Line 29 at page 3, is this a type?*

Response:

We appreciate the reviewer for noticing this typo. Correct, it is NCS.1.2 and NCS.1.3 that show low binding to A/Wisconsin/67/2005 N2 recombinant NA protein. We have corrected this mistake.

6. *Lines 8-10 at page 14, is the figure legend of Fig. 1d and Fig. 1e inversely indicated?*

Response:

We thank the reviewer for catching this mistake. We have corrected this in our revised manuscript.

7. *Is there high-resolution data for NCS.1.1-, NCS.1.2-, and NCS.1.3-bound NA of B/Vic CO17 (Fig.4g)?*

Response:

At present, we do not have high-resolution cryoEM structures of NCS.1.x Fabs bound to B/Vic CO17. However, our binding and functional data, along with the nsEM characterization in the manuscript, support that these antibodies engage B/Vic NA in a similar fashion. Given these existing data, we do not believe that additional confirmatory high-resolution structures are necessary, although such structures may provide more comprehensive structural understanding on how the antibodies tolerate variations at positions including 247 and 343 as well as the lack of glycosylation at position 273. We have included a discussion of potential future structural studies to further explore this interaction in the revised manuscript.

Page 9, lines 9–12:

“Future high-resolution cryoEM studies of NCS.1.x mAbs bound to additional NA subtypes, including B/Vic CO17, could further clarify the role of water-mediated interactions in broad NA targeting. Similarly, studies of other catalytic site-targeting mAbs may offer additional insights if water molecules are explicitly modeled.”

Reviewer #3 (Remarks to the Author):

Lederhofer, Borst et al., report here on monoclonal antibodies (mAbs) that target the catalytic site (NCS) of the influenza virus receptor-destroying enzyme, neuraminidase.

The authors isolated, and comprehensively characterize these mAbs and showed that they were broadly against several NAs, including N1, N5, and B NA. Mostly notably, the cryo-EM structures of NCS mAbs in complex with NAs revealed that the amino acid residues at the tips of the mAbs' CDR3 loops mimic the enzyme's substrate, sialic acid. Given the relatively high resolution of their cryo-Em reconstructions, the authors were able to identify a network of water molecules involved in the solvation of the antibody epitope and arranged in a similar fashion to that of SIA-bound NA structure.

This is a high-quality manuscript and my suggestions on improvement are minor:

Response:

We thank the reviewer for the positive notes. Our detailed response to specific comments follow.

1 - *It appears that many, if not all, glycans have issues with stereochemistry. I suggest using Privateer to validate and correct the conformers.*

Response:

We appreciate this suggestion and have now used Privateer to validate glycan stereochemistry. We identified and corrected several minor deviations in glycan conformers, ensuring compliance with expected stereochemistry. The updated structural models reflect these refinements, and we have included a statement in the Methods describing the validation process and added a relevant reference (ref. 67).

2 - Why are hydrogens included in the final atomic coordinates? The resolution does not allow to visualize them, so they should only be used in the refinement steps to minimize clashes, but removed from the final model.

Response:

We totally agree with the reviewer. Hydrogens were included in the model during refinement for clash minimization but were inadvertently retained in the final coordinates. We have now removed all hydrogen atoms from the deposited structures, in line with standard cryoEM modeling practices.

3 - Could you elaborate on the cryo-EM data processing? It looks like the water molecules were overzealously placed, i.e., many are not visible at medium and higher map contours. Did you use phenix.douse, for example, to check? While on that point, I wonder how many waters get averaged out simply by enforcing the C4 symmetry? Are they all also present in the C1 refined structure?

Response:

We thank the reviewer for raising this important point regarding water placement and validation. In response, we undertook additional steps to rigorously evaluate the accuracy and consistency of placed water molecules across our maps, and we have added a new section in the Methods to describe this process in detail.

Page 27–28, lines 34–19:

“CryoEM water placement and validation

Initial water molecules were placed into cryoEM density using Phenix.douse on the C4 symmetry–refined map of the NCS.1.1/N1-sNAp complex, followed by manual validation and adjustment in Coot. Additional water molecules were manually added in regions where unmodeled density could be reasonably attributed to water. These placements were further refined through molecular dynamics with flexible fitting in ISOLDE, allowing the water molecules to relax into energetically favorable positions that better matched the local density.

To validate these placements, the structure from the C4 map was docked into a C1 symmetry–refined map of the same complex, and water positions were reassessed using CheckWaters in ChimeraX. A density threshold of 0.2 was applied in ChimeraX, meaning that waters were only retained if they were supported by map values at or above this contour level.

Waters below this threshold were removed (165 in total), as they were likely attributable to noise or weak occupancy averaged out in the C4 map. For the C4 map, waters that were retained following these steps were kept for the C4-refined PDB structure.

For the C1 map, Phenix.douse was rerun with a deliberately permissive map_threshold_scale of 0.7 and sphericity_filter set to false to ensure comprehensive sampling of potential water sites. The resolution parameter was set to 2.5 Å, consistent with the global resolution of the map (with local resolution at the NCS.1.1–N1 interface being higher). It is important to note that the Phenix.douse threshold governs initial water placement, while the ChimeraX threshold is used post hoc to further assess whether placed waters are supported by density.

Following automated placement in the C1 map, manual inspection was conducted for all water molecules, with particular focus on those at the NCS.1.1–N1 interface. Water placements were also compared across all protomers in the C1 asymmetric unit, revealing that many waters were consistently found in the same positions across symmetry-independent copies. This internal consistency, along with strong map support and high resolution, provides high confidence that the retained waters represent genuine structural features rather than noise.”

4 - The methods section could be more detailed, especially with NA purification etc. For example, Ca²⁺ is present in the structure but there is no mention of calcium in the purification or grid preparation protocols.

Response:

We have now revised the Methods section to include a detailed description of NA purification, including the use of Ca²⁺ in buffer conditions where applicable. The presence of Ca²⁺ in the cryoEM structures is consistent with prior NA structural studies, and we have clarified that Ca²⁺ was only present in the culture medium—no additional Ca²⁺ was included during purification. We described how we purify recombinant NA in our previous paper (Ellis and Lederhofer et al. *Nat Commun*, 2022) and the same protocol was used for this manuscript. We have clarified this point in the Methods.

Page 23, lines 31–36:

“Recombinant NA proteins were expressed by transient transfection of expression vectors in Expi293 cells using ExpiFectamine 293 Transfection Kit (ThermoFisher Scientific). Proteins were purified from cell culture supernatants five days post-transfection. After immobilized metal affinity chromatography by Ni-Sepharose, proteins were further purified by size exclusion chromatography into phosphate-buffer saline (PBS) as described previously¹⁸. No additional Ca²⁺ was added during protein expression and/or purification.”

5 - You could consider IMGT numbering for the mAbs, instead of Kabat.

Response:

We thank the reviewer for their suggestion using IMGT numbering for the mAbs. Since the Kabat numbering scheme is as commonly used as IMGT numbering in literature, we decided to keep the Kabat numbering to be consistent with our previous papers.

6 - (d) and (e) are inverted in Figure 1 legend

Response:

We thank the reviewer for catching this mistake. We have corrected this in our manuscript.

P.S. Thank you for sending the primary cryo-EM maps - that made my job much easier!

Response:

We thank the reviewer for their kind comment and are glad that the provided cryoEM maps and models were helpful.